# Membrane voltage dysregulation driven by metabolic dysfunction underlies bactericidal activity of aminoglycosides

Giancarlo Noe Bruni, Joel M Kralj*

BioFrontiers Institute and the Department of Molecular, Cellular, Developmental Biology, University of Colorado Boulder, Boulder, United States

**Abstract** Aminoglycosides are broad-spectrum antibiotics whose mechanism of action is under debate. It is widely accepted that membrane voltage potentiates aminoglycoside activity, which is ascribed to voltage-dependent drug uptake. In this paper, we measured the response of *Escherichia coli* treated with aminoglycosides and discovered that the bactericidal action arises not from the downstream effects of voltage-dependent drug uptake, but rather directly from dysregulated membrane potential. In the absence of voltage, aminoglycosides are taken into cells and exert bacteriostatic effects by inhibiting translation. However, cell killing was immediate upon re-polarization. The hyperpolarization arose from altered ATP flux, which induced a reversal of the F1Fo-ATPase to hydrolyze ATP and generated the deleterious voltage. Heterologous expression of an ATPase inhibitor completely eliminated bactericidal activity, while loss of the F-ATPase reduced the electrophysiological response to aminoglycosides. Our data support a model of voltage-induced death, and separates aminoglycoside bacteriostasis and bactericide in *E. coli*.

## Introduction

Aminoglycosides are a potent class of translation inhibitor antibiotics with a broad activity spectrum. Despite a long history in the clinic (*Krause et al., 2016*), their exact mechanism of action remains unclear (*Ezraty et al., 2013*; *Keren et al., 2013*; *Kohanski et al., 2007*). In Gram-negative bacteria, aminoglycosides must cross the outer membrane and plasma membrane (*Taber et al., 1987*), into the cytoplasm where they can exert their bactericidal effect which requires binding to the ribosome (*Davis, 1987*). The kinetics of uptake into the cytoplasm have been extensively studied and occur in three steps (*Taber et al., 1987*). An ionic interaction between the polycationic aminoglycosides and the outer membrane of the bacterial cell induces a disruption of the outer membrane (*Hancock et al., 1981*), and allows the aminoglycoside to ionically associate with the inner membrane (*Bryan and Van Den Elzen, 1977*). The next step is known as the energy-dependent phase I (EDP-I) and occurs almost instantaneously upon aminoglycoside treatment (*Muir et al., 1984*). This portion is noted as energy dependent because both respiration inhibitors (*Leviton et al., 1995*) and differential carbon sources (*Nichols and Young, 1985*) reduced uptake. EDP-I is thought to be the step at which the aminoglycoside enters the cytoplasm (*Taber et al., 1987*; *Nichols and Young, 1985*), is concentration dependent (*Bryan and Van Den Elzen, 1977*), and occurs in cells that are resistant to or tolerant of aminoglycosides (*Ezraty et al., 2013*; *Bryan and Van den Elzen, 1976*). Following EDP-I is EDP-II, which only occurs in aminoglycoside-sensitive cells (*Taber et al., 1987*; *Bryan and Van den Elzen, 1976*), is thought to be essential for the bactericidal activity of aminoglycosides, and requires respiration (*Bryan and Van den Elzen, 1976*). Throughout these early studies, uptake of the aminoglycosides was often treated as synonymous with bactericidal activity.

Proposed bactericidal mechanisms all stem from this consensus theory of aminoglycoside uptake (*Ezraty et al., 2013*; *Kohanski et al., 2007*; *Leviton et al., 1995*; *Kohanski et al., 2008*;

*For correspondence:
joel.kralj@colorado.edu

Competing interests: The authors declare that no competing interests exist.

*Davis et al., 1986*). Once aminoglycosides are inside the cell, several competing theories exist to explain bactericidal activity including membrane breakdown from mistranslated protein (*Davis et al., 1986*; *Busse et al., 1992*), reactive oxygen species (*Kohanski et al., 2007*) (ROS), and a positive feedback of drug uptake (*Ezraty et al., 2013*; *Leviton et al., 1995*), although there is debate around each (*Ezraty et al., 2013*; *Kohanski et al., 2008*; *Fraimow et al., 1991*). Despite this debate, there is broad agreement upon two important points. The first is that the uptake mechanism, and therefore the resulting bactericidal activity, is voltage dependent (*Damper and Epstein, 1981*). That is, bactericidal activity occurs after uptake, and that uptake is intrinsically tied to membrane potential (*Ezraty et al., 2013*; *Taber et al., 1987*; *Davis et al., 1986*). This makes sense given the ample evidence of broken respiration protecting bacteria from aminoglycosides (*Ezraty et al., 2013*; *Nichols and Young, 1985*; *Lobritz et al., 2015*; *McCollister et al., 2011*). The second point is that this voltage induced uptake is responsible for mistranslation of protein upon aminoglycoside binding, which in turn creates the membrane breakdown essential for bactericidal activity. These pores, or the ROS produced in their occurrence, are thought to be responsible for the bactericidal activity of aminoglycosides (*Davis et al., 1986*; *Kohanski et al., 2008*). New techniques offer the ability to study the effects of aminoglycosides and perhaps resolve some debated aspects of their mechanism of action.

Single cell, fluorescent imaging offers a means to shed light on the effects of antibiotic exposure with high resolution in space and time. Improvements in microscope hardware enable automated live cell imaging while resolving the responses of individual bacteria. This hardware can be coupled with genetically encoded, or chemical fluorescent sensors that report bacterial voltage (*Kralj et al., 2011*; *Prindle et al., 2015*; *Stratford et al., 2019*), calcium (*Bruni et al., 2017*), and ATP (*Tantama et al., 2013*; *Yaginuma et al., 2015*), providing a lens to explore the long-term effects of antibiotic exposure. Recently, live cell voltage imaging of *Bacillus subtilis* revealed the importance of membrane potential in response to translation inhibitors (*Lee et al., 2019*). These new tools highlight the importance of membrane potential controlling bacterial physiology, and our ability to now study electrophysiology at the single-cell level.

Despite the debate on the bactericidal mechanism of aminoglycosides, there is broad agreement that bacterial membrane potential plays a critical role. In this paper, we sought to investigate the influence of membrane potential in mediating bactericide upon treatment with aminoglycosides. We used live cell microscopy to maintain high spatial and temporal resolution while also resolving any heterogeneity within the population. We found that lethal concentrations of aminoglycosides-induced voltage hyperpolarization leading to large fluctuations in cytoplasmic calcium that persisted for >48 hr after treatment. We found these transients were correlated with the inability of cells to regrow, giving us a technique to measure the onset of cell death in real time at the single-cell level. We found evidence that the transients arise from decreased ribosomal consumption of ATP leading to a reversal of the F1Fo-ATPase. The voltage hyperpolarization, in tandem with mistranslated proteins in the membrane, induced the bactericidal action. Our model proposes a new mechanism which links the chemical energy state of the cell with membrane potential dysregulation that can lead to death.

## Results

### Voltage is not necessary for aminoglycoside uptake or inner membrane pore formation in *E. coli* but is required for bactericidal activity

The proton ionophore cyanide m-chlorophenyl hydrazine (CCCP) dissipates voltage gradients, and is known to protect *E. coli* against the bactericidal activity and EDP-II uptake of aminoglycosides (*Taber et al., 1987*; *Davis, 1987*). A colony-forming unit (CFU) assay was performed using a glucose minimal medium (PMM, see Materials and method) in the presence of aminoglycosides. These measurements showed cells continued to grow in PMM in the presence or absence of CCCP (*Figure 1A*). Treatment of cells with aminoglycosides alone caused a rapid reduction in CFUs. In contrast aminoglycoside treatment of cells pre-treated with CCCP showed bacteriostatic activity (*Figure 1A*).

To more carefully examine the contrasting data that CCCP-treated cells were growth inhibited in the presence of aminoglycoside, and the evidence that voltage is necessary for aminoglycoside uptake, a polysome analysis was used to assess ribosomal assembly in these conditions

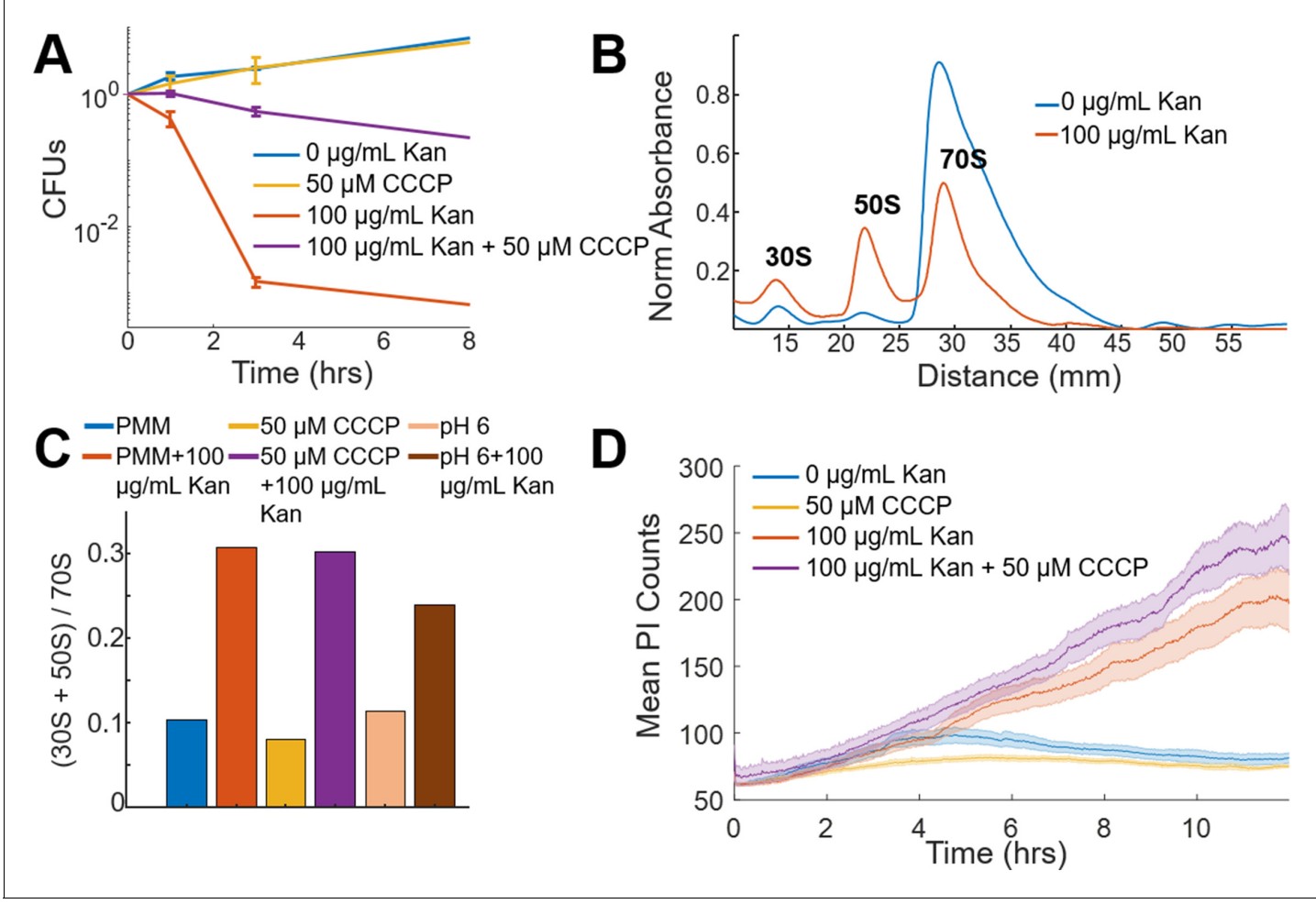

**Figure 1.** Voltage is not necessary for aminoglycoside uptake or inner membrane pore formation in *E. coli* but is required for bactericidal activity. (A) Colony forming units (CFUs) of untreated cells (blue) over four time points compared to cells treated with 50 µM CCCP (yellow), 100 µg/mL kanamycin (orange), and 50 µM CCCP + 100 µg/mL kanamycin (purple). Each curve averages three biological replicates, with mean and standard deviation plotted for each time point. (B) Ribosomal sucrose gradient depth plotted against 254 nm absorbance from LB grown *E. coli* treated with vehicle (blue), 100 µg/mL kanamycin (orange). The 30S, 50S, and 70S peaks are labeled. (C) Ratio of the area under the curve for the 30S + 50S to 70S peaks from *E. coli* in PMM pH 7.5, +50 µM CCCP, or pH 6 in the presence or absence of kanamycin. (D) Propidium iodide (3.75 µM in PMM) fluorescence in cells that were untreated (blue), 50 µM CCCP (yellow), 100 µg/mL kanamycin (orange), and 50 µM CCCP + 100 µg/mL kanamycin (purple) treated. The curve is the mean (solid) and standard deviation (shaded) for three biological replicates.

The online version of this article includes the following figure supplement(s) for figure 1:

**Figure supplement 1.** Aminoglycosides enter cells and induce ribosomal dissociation in the abscence of membrane voltage.

(*Figure 1B*; *Qin and Fredrick, 2013*). Untreated cells showed a majority of 70S particles, while addition of aminoglycosides caused a large fraction of ribosomes to split into 30S and 50S subunits (*Zhang et al., 2015*). Unexpectedly, ribosomes in aminoglycoside-treated cells showed equal dissociation in the presence or absence of CCCP (*Figure 1C*, *Figure 1—figure supplement 1*), despite the dramatic difference in drug activity. Aminoglycoside treatment at pH 6, which also has reduced membrane potential (see Materials and methods), showed bacteriostatic activity and ribosomal dissociation (*Figure 1C*). In addition to chemical perturbations, naturally occurring mutations in bacterial populations can lead to protection against aminoglycosides arising from a decrease in membrane potential (*Ezraty et al., 2013*; *Damper and Epstein, 1981*). These mutations often occur in the electron transport chain and reduce aminoglycoside uptake while concomitantly increasing survival (*Ezraty et al., 2013*). Mutations of genes in the *nuo* operon have reduced uptake and death (*Ezraty et al., 2013*), but have equivalent aminoglycoside-induced ribosomal dissociation (*Figure 1—*

*figure supplement 1B*). Although uptake of aminoglycosides in the absence of membrane potential has been observed (*Fraimow et al., 1991*), the equivalent effect on ribosomal fraction abundance in *E. coli,* independent of voltage, had not been observed previously to our knowledge.

The clear uptake of aminoglycosides in the absence or alteration of membrane voltage suggested mistranslated proteins that induce membrane pores (*Kohanski et al., 2008*; *Davis et al., 1986*) could also occur. We measured the uptake of propidium iodide (PI), a membrane-impermeable DNA-binding fluorescent dye, in the presence of aminoglycosides. The aminoglycoside-treated population showed increasing PI fluorescence as compared to untreated cells (*Figure 1D*), indicating a loss of membrane integrity which correlated with the kinetics of cell death when measured by CFUs. Pre-treating cells with CCCP, however, showed a similar aminoglycoside-induced increase in PI fluorescence, despite the switch from bactericidal to bacteriostatic activity. Chloramphenicol, a bacteriostatic translation inhibitor, induced only small increases in PI fluorescence (*Figure 1—figure supplement 1C*). Fluorescently labeled gentamicin texas-red (GTTR) also showed an increase in concentration in the presence or absence of CCCP (*Figure 1—figure supplement 1D*), although the increases after 1 hr could be due to a destabilized membrane, similar to the results with PI. These data suggested that protein mistranslation and membrane destabilization occur in the absence of membrane potential and are not sufficient to cause bactericidal activity. Given the discrepancy between CFUs, ribosomal dissociation, and PI uptake, we hypothesized voltage led to bactericide through mechanisms other than drug uptake. We therefore considered if bactericidal activity could arise through a combination of the mistranslated protein-induced pore formation and membrane hyperpolarization. In order to test this hypothesis, we turned to single-cell measurements of bacterial electrophysiology.

## Voltage and calcium exhibit altered electrophysiological flux in response to aminoglycosides

Fluorescent sensors of voltage and calcium have been used to monitor electrophysiology in bacteria at the single-cell level with high time resolution (*Stratford et al., 2019*; *Bruni et al., 2017*; *Lee et al., 2019*; *Sirec et al., 2019*). We used the genetically encoded sensor, PROPS, to measure voltage dynamics after 2 hr of treatment with kanamycin. The aminoglycoside-treated cells had larger fluorescent transients as compared to untreated cells (*Figure 2—figure supplement 1A*), but the high light intensities required prohibited long-term monitoring of single cells. GCaMP6, a fluorescent calcium indicator, is bright and sensitive enough to monitor live cells over hours or days, and we previously established calcium spikes were intrinsically linked to voltage fluctuations (*Bruni et al., 2017*). Individual *E. coli* expressing a fusion of GCaMP6f (calcium sensor) and mScarlet (spectrally independent control) were imaged upon exposure to 0 µg/mL or 100 µg/mL kanamycin and were monitored for 8 hr. Cells treated with antibiotic ceased growth and after ~2 hr showed large, non-oscillatory fluctuations which were uncoordinated between neighboring cells and not seen in untreated cells (*Figure 2A*, *Video 1*). Untreated *E. coli* had few cells that exhibited transients compared to drug-treated cells, and untreated cells grew and divided normally which indicated the transients were not a phototoxic effect (*Figure 2—figure supplement 1B,C*, *Video 2*). These drug-induced transients were larger than previously observed mechanically induced fluctuations (*Bruni et al., 2017*). At a concentration of 30 µg/mL kanamycin >99.99% of cells cannot form colonies after 6 hr, yet we saw transients > 48 hr after kanamycin treatment at that concentration (*Figure 2—figure supplement 2D,E*). The delay between antibiotic exposure and the appearance of calcium transients varied across the population with a mean time of 1.64 hr after treatment (*Figure 2—figure supplement 2F*). The fraction of cells showing transients increased with increasing concentrations of kanamycin (*Figure 2B*). These data showed that aminoglycosides induced large electrophysiological effects that arise at similar timescales to cell death measured by CFUs.

In order to compare the kinetics of the aminoglycoside response of populations of cells across treatment conditions, we needed a metric that would encompass the fluorescent dynamics across many cells. To visualize the transients across a population, a moving standard deviation was calculated for each cell, and then averaged across all cells. This mean of the moving standard deviation (taken from 30 to 500 cells) was considered one biological replicate, and the average and standard deviation of three biological replicates is then plotted (*Figure 2C*, *Figure 2—figure supplement 2*). This metric will depend strongly on the microscope system used, and thus requires relative comparisons of treated versus control under otherwise identical imaging conditions. We defined a drug-

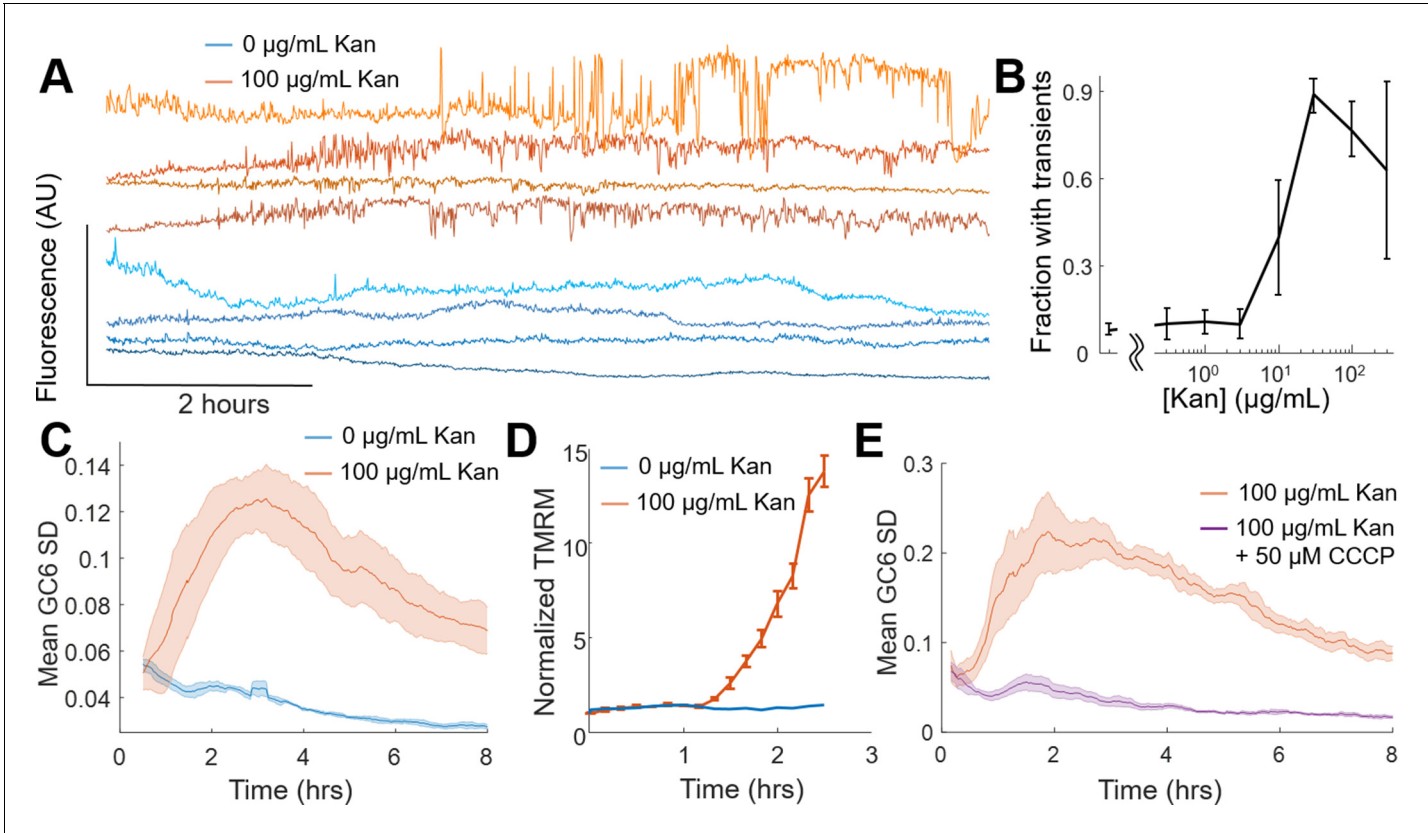

**Figure 2.** Voltage and calcium exhibit altered electrophysiological flux in response to aminoglycosides. (**A**) Time traces of GCaMP6 fluorescence from single cells treated with 0 μg/mL (blue shades) and 100 μg/mL (orange shades) kanamycin. Individual cells display non-oscillatory transients. (**B**) The fraction of cells in a population of GCaMP6F expressing cells *E. coli* experiencing the transients in A at different concentrations of Kanamycin. The mean (line) and standard deviation (error bars) are shown for three biological replicates. (**C**) The average (solid line) and standard deviation (shading) of the moving GCaMP6f standard deviation (SD) over time from 0 μg/mL (blue) and 100 μg/mL kanamycin (orange) treated cells. (**D**) TMRM fluorescence from untreated (blue) or kanamycin treated (orange) (100 μg/mL, 2 hr) *E. coli* measured by cytometry. The average (line) and standard deviation (error bars) of three biological replicates are plotted. (**E**) The average (solid line) and standard deviation (shading) of the moving GCaMP6f Standard Deviation over time from 100 μg/mL kanamycin-treated cells in the absence (orange) or presence (purple) of 50 μM CCCP.

The online version of this article includes the following figure supplement(s) for figure 2:

**Figure supplement 1.** Kanamycin induces voltage and calcium transients.
**Figure supplement 2.** Calculating moving standard deviation from treated cells.
**Figure supplement 3.** Only aminoglycosides induce calcium transients.
**Figure supplement 4.** High pH is necessary to induce calcium transients.

induced calcium transient as any cell that showed a moving standard deviation (SD) >7 fold above untreated cells for >40 min. The GCaMP moving SD metric can separate treated and untreated populations of *E. coli*. All aminoglycosides tested exhibited a concentration-dependent onset of calcium transients, as well as significantly increased GCaMP SD, but other bacteriostatic or bactericidal antibiotics had neither (*Figure 2—figure supplement 3A,B*). Our measurements do not rule out the possibility of other ions moving across the membrane (*Dubin and Davis, 1961*), and indeed we see that proton concentrations as measured by the red fluorescent pH indicator, pHuji (*Shen et al., 2014*) also show transients, but their initial amplitude is much smaller than the calcium transients (*Figure 2—figure supplement 3C,D*). A lack of sufficient sensors prohibited us from measuring other ions at these temporal and spatial scales.

Given the observation that CCCP and low pH eliminated the calcium transients, we hypothesized that these large fluorescent changes were a product of a more polarized membrane potential, which would be consistent with the positive feedback of drug uptake model (*Ezraty et al., 2013*; *Bryan and Van Den Elzen, 1977*; *Davis et al., 1986*). Tetramethylrhodamine methylester (TMRM), a

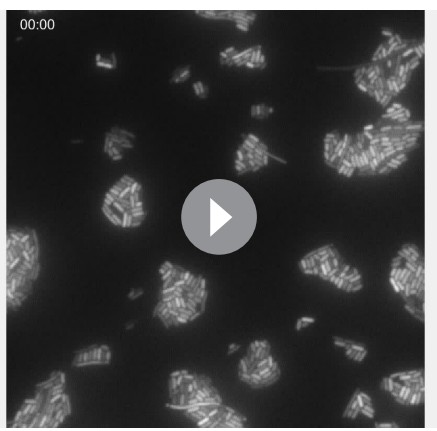

**Video 1.** Video of *E. coli* expressing GCaMP6f-mScarlet upon treatment with 100 µg/mL kanamycin. The movie was taken using 488 nm excitation and a 40x air objective imaged onto an sCMOS camera. The movie was taken at a sampling rate of 1 image per minute for 16 hr. This movie has been corrected for uneven illumination, XY drift, and background as mentioned in the Materials and methods. The time indicated represents HH:MM.
https://elifesciences.org/articles/58706#video1

membrane permeable fluorescent voltage reporter, accumulates in polarized mitochondria (*Zorova et al., 2018*) and *E. coli* (*Kralj et al., 2011*; *Lo et al., 2007*). Untreated *E. coli* showed no change in intracellular TMRM levels over 2.5 hr (*Figure 2D*). Cells treated with kanamycin showed a sharp increase in TMRM fluorescence after 80 min, corresponding to a change of −72 mV after 2.5 hr (see Materials and methods). Assuming a resting potential of −150 mV the treated cells would have a membrane voltage of −222 mV. This observation is consistent with an aminoglycoside-induced change in membrane potential occurring at the same time as the calcium transients.

If aberrant voltage induced the calcium transients, dissipating the voltage would eliminate the transients. Cells expressing GCaMP6 were treated with CCCP and compared to kanamycin exposure alone (*Figure 2E*). CCCP-treated cells showed no increase in GCaMP6f SD, or individual calcium transients. Cells treated at pH 6 also showed no increase in calcium transients (*Figure 2—figure supplement 4A*) and showed no hyperpolarization measured by TMRM (*Figure 2—figure supplement 4B*). Knockouts of the *nuo* operon show altered kinetics in the onset of the GCaMP6f SD, as well as a lower amplitude in response to aminoglycoside treatment (*Figure 2—figure supplement 4C*). Together, these data show that aminoglycosides-induced hyperpolarization and large ionic fluctuations only in the presence of membrane voltage, and that chemical or genetic alterations of membrane voltage affect the GCaMP6 response.

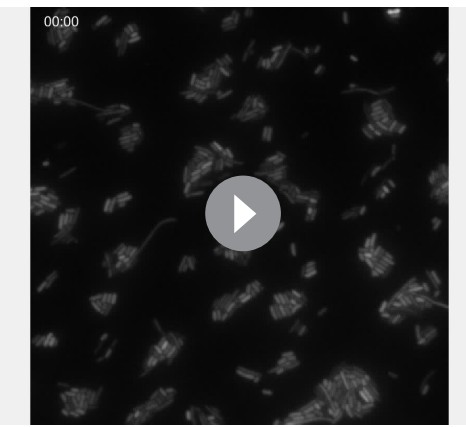

**Video 2.** Video of *E. coli* expressing GCaMP6f-mScarlet with no kanamycin addition. The video was taken using 488 nm excitation and a 40x air objective imaged onto an sCMOS camera. The movie was taken at a sampling rate of 1 image per minute for 16 hr. This movie has been corrected for uneven illumination, XY drift, and background as mentioned in the Materials and methods. The time indicated represents HH:MM.
https://elifesciences.org/articles/58706#video2

## Single-cell calcium flux predicts cellular aminoglycoside response

The onset of voltage hyperpolarization, calcium transients, and cell death as measured by CFUs suggested the observed fluorescent calcium traces could be a good technique to measure bactericide at the single-cell level. Fluorescence measurements were taken under continuous flow during the addition, then removal, of kanamycin. As expected, antibiotic exposure induced large calcium transients in many cells. After 4 hr of kanamycin exposure, medium without drug was added, and ~2% of cells reinitiated cell division (recovered cells, 35/1727 cells, *Figure 3A*, *Video 3*). Of the 35 recovered cells, none exhibited drug-induced calcium transients during or after antibiotic exposure (*Figure 3B*), and the population of recovered cells had lower calcium fluctuations as compared to arrested cells (*Figure 3C*). Recovered cells were not genetically resistant, as a second exposure to kanamycin stopped growth and induced calcium transients in daughter cells (*Figure 3—figure supplement 1A–C*). Finally, within an untreated

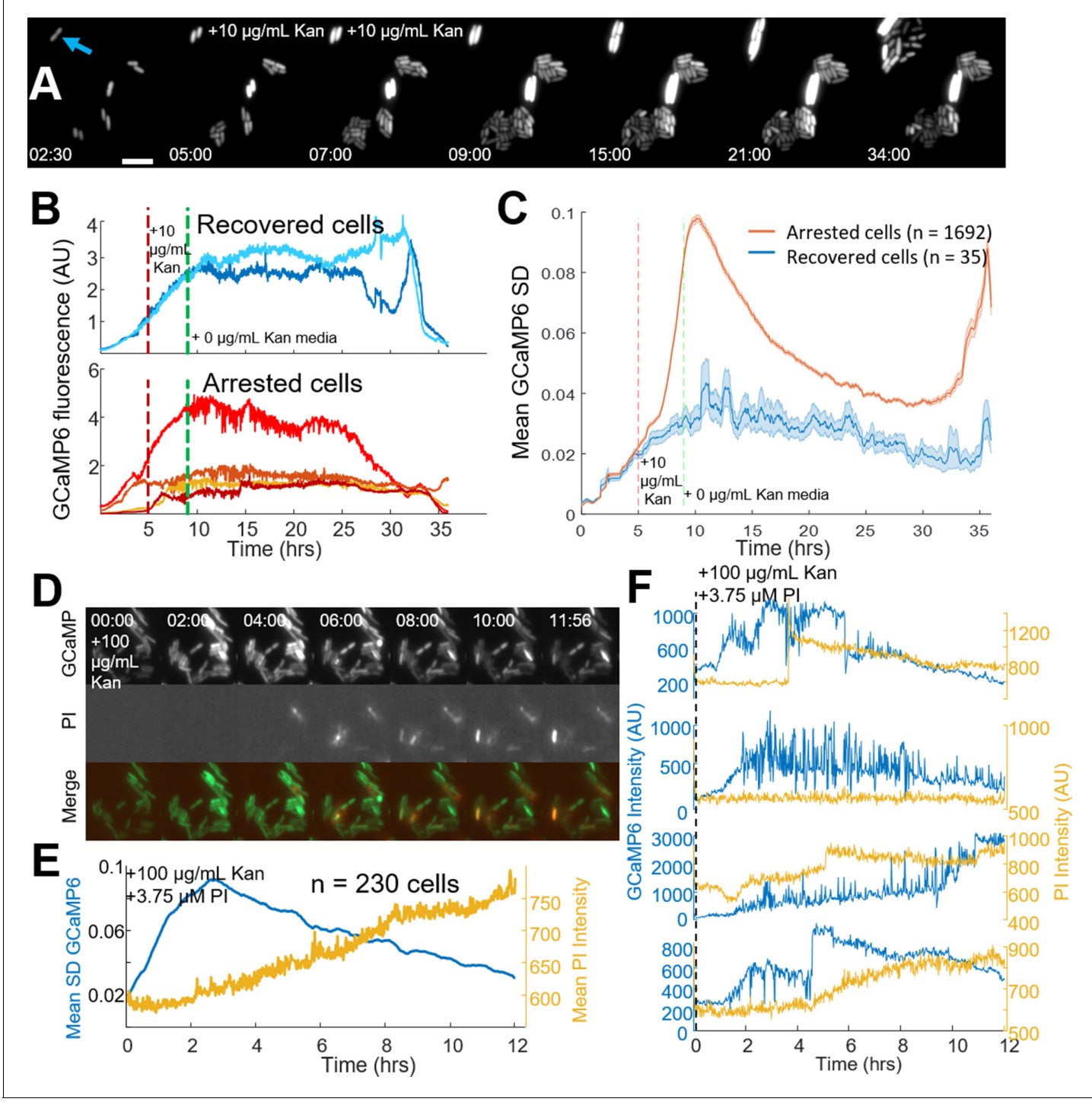

**Figure 3.** Single-cell calcium flux predicts cellular aminoglycoside response. (**A**) Strip chart of cells expressing GCaMP6f. Cells were imaged in PMM alone for 5 hr, then exposed to 10 µg/mL kanamycin for 4 hr. After 4 hr, PMM alone was flowed in for an additional 26 hr. The blue arrow indicates a cell that was able to divide after treatment with kanamycin, 5 µm scale bar. Time is shown in (HH:MM) format. (**B**) Individual GCaMP6 time traces from cells that regrow after treatment compared to a random selection of cells that do not regrow within 24 hr. (**C**) The average (line) and standard deviation (shaded region) of the moving SD from all cells that regrow (blue) vs those that do not regrow (red). (**D**) Strip chart showing GCaMP6 fluorescence (top), propidium iodide fluorescence (middle), and the merge. Cells were treated with 100 µg/mL kanamycin at time t = 0. Time is shown in (HH:MM) format. (**E**) The mean GCaMP6 standard deviation for the population is shown in blue. Yellow shows the population average of the PI fluorescence. (**F**) Time traces of individual cells showing the GCaMP6 fluorescence (blue) and the PI fluorescence (yellow) on the same cells. The PI fluorescence was not correlated with the onset of transients, although many cells did uptake PI during the course of the experiment.

*Figure 3 continued on next page*

*Figure 3 continued*

The online version of this article includes the following figure supplement(s) for figure 3:

**Figure supplement 1.** Cells that did not experience calcium transients are not genetically resistant.

population, a small fraction of cells exhibited transients (22 of 1544), where each cell with transients did not divide (*Figure 3—figure supplement 1D–F*). In all cases, tested calcium transients correlated with reduced population viability; conditions with fewer calcium transients increased CFUs, and any cell that exhibited transients did not regrow. This data provided a technique to measure one hallmark of single-cell death in *E. coli* in real time as all observations of these transients indicated that a cell experiencing them was rendered unable to divide, although we are not able to definitively say that the transients caused cell death.

Spectrally separating PI and GCaMP enabled us to study the kinetics between catastrophic calcium transients and pore formation in single cells. The mistranslation that causes pore formation was previously measured to occur within a half hour of aminoglycoside treatment (*Davis et al., 1986*). We hypothesized that mistranslated proteins in the plasma membrane created an ionic imbalance in polarized cells leading to the observed calcium transients. To test our hypothesis, we incubated GCaMP6 expressing *E. coli* with PI in the presence of aminoglycoside (*Figure 3D*). The population average showed a smoothly increasing level of PI uptake upon aminoglycoside exposure (*Figure 3E*), similar to our earlier data. However, the GCaMP6 moving SD increased well before appreciable PI uptake. Individual cells showed calcium transients preceded PI entry into the cytoplasm, and that PI often increased in very large bursts (*Figure 3F*). Thus, pores large enough to accommodate PI occurred after aminoglycoside-induced hyperpolarization and catastrophic calcium transients, suggesting bactericidal activity occurred prior to pore formation.

## Voltage toggles between bactericidal and bacteriostatic activity in aminoglycoside-treated cells

The data above showed that aminoglycoside uptake, ribosome dissociation, and mistranslated protein can occur without membrane potential. Aminoglycosides in the absence of a voltage exhibited a bacteriostatic effect, but voltage induced bactericide. We therefore sought to explore the requirements of voltage as the bactericidal keystone in *E. coli* by using the calcium transients as a real time marker of permanent cell cycle arrest, while controlling the chemical environment to actuate membrane voltage.

Treating cells with aminoglycoside-induced calcium transients (*Figure 4A* top, *Figure 4—figure supplement 1A* top) as expected. However, removing the voltage either through addition of CCCP or lowering pH immediately ceased all transients at the single cell and population levels (*Figure 4A,B*, *Figure 4—figure supplement 1A, B*), although no cells re-initiated cell division. Thus, voltage was necessary for the calcium transients to occur. Conversely, *E. coli* was incubated with kanamycin in the presence or absence of CCCP for 4 hr and showed calcium transients only in the cells without CCCP as expected (*Figure 4C* top). Removal of kanamycin and CCCP initiated transients within 7 min, much faster than the appearance of transients from aminoglycoside treatment without CCCP (*Figure 4C,D*). Similar results were seen exchanging pH 6 with pH 7.5 to reestablish a membrane voltage (*Figure 4—figure supplement 1C,D*). The rapid onset showed that aminoglycosides can exert bactericidal activity immediately upon reestablishment of membrane

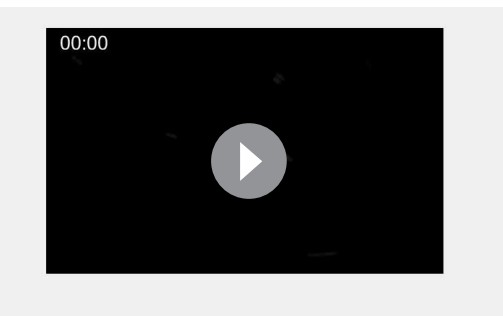

**Video 3.** Video of *E. coli* expressing GCaMP6f-mScarlet switching the medium from PMM (0–5 hr), PMM + 10 μg/mL kanamycin (5–9 hr), PMM (9–35 hr). The movie was taken at a sampling rate of 1 image per minute for 29 hr. This movie has been corrected for uneven illumination, XY drift, and background as mentioned in the Materials and methods. The time indicated represents HH:MM.

https://elifesciences.org/articles/58706#video3

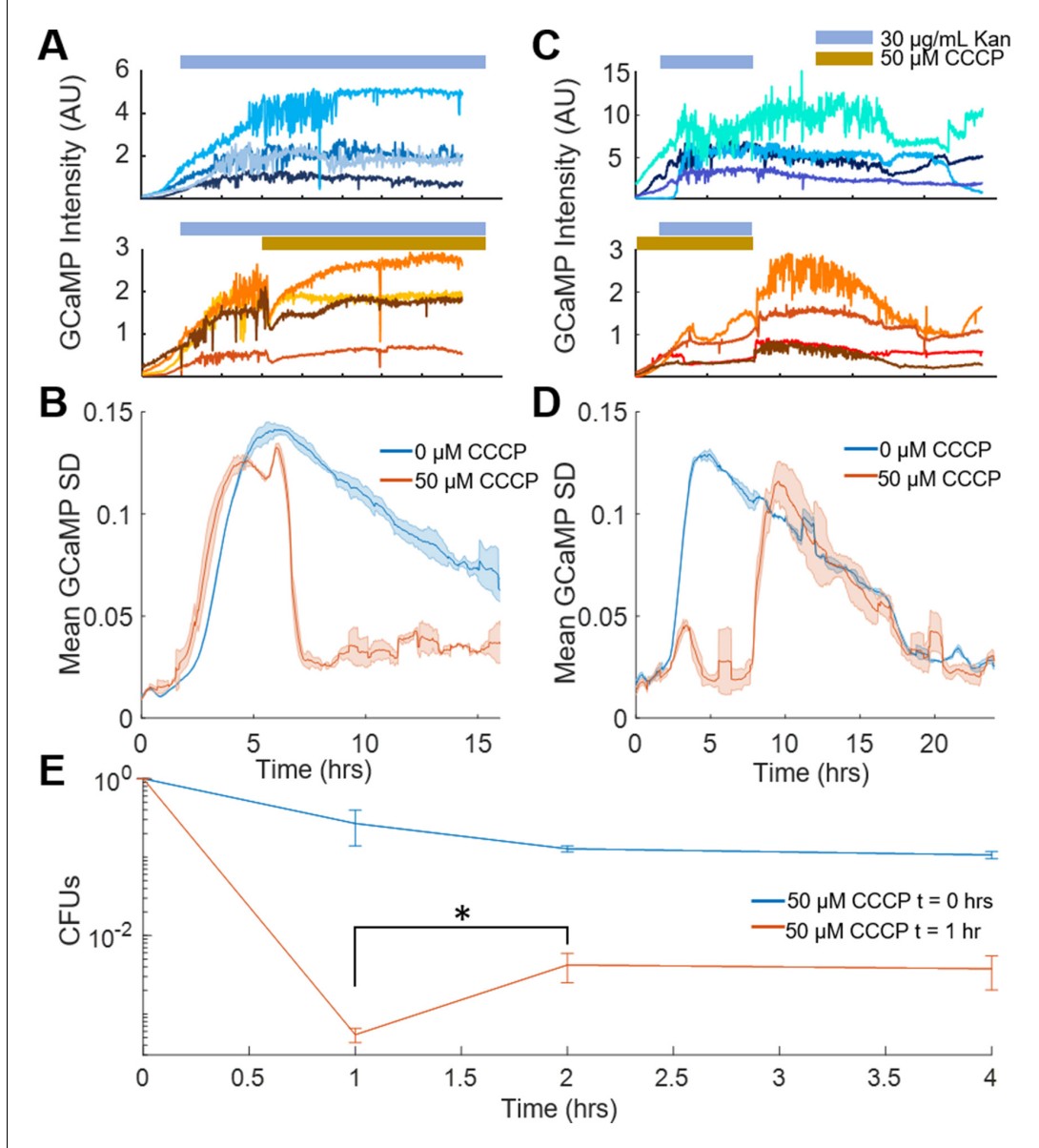

**Figure 4.** Voltage toggles between bactericidal and bacteriostatic activity in aminoglycoside-treated cells. (A) Single-cell traces of GCaMP6f intensity over time upon treatment with kanamycin (blue bar, top), or with kanamycin followed by CCCP (yellow bar, bottom). (B) Mean GCaMP6f moving SD of biological replicates over time. The population traces are the mean of the single-cell experiments in B, with kanamycin (blue, corresponds to A top) or kanamycin +CCCP-treated cells (orange, corresponds to A bottom). (C) Single-cell traces of GCaMP6f intensity over time after kanamycin (indicated with a blue bar) was flowed across the cells at 2 hr that were pretreated with vehicle (top) or CCCP (bottom, indicated with a yellow bar). Kanamycin and CCCP were then flowed out of the chamber at 6 hr. (D) Mean GCaMP6f moving SD of biological replicates over time. The population traces are the mean of the single-cell experiments in C, with kanamycin (blue, corresponds to C top) or kanamycin +CCCP-treated cells (orange, corresponds to C bottom). (E) Cells treated with 10 μg/mL gentamycin with CCCP added at t = 0 hr (blue line), or CCCP added at t = 1 hr (orange line). CFUs taken at 1 hr were counted before addition of CCCP, which increased the number of surviving cells at t = 2 hr. *p<0.05.

The online version of this article includes the following figure supplement(s) for figure 4:

**Figure supplement 1.** Catastrophic calcium transients require a membrane potential as indicated by low pH protection.

voltage, and that in the conditions tested, voltage is sufficient to induce catastrophic calcium transients which were correlated with cell death.

If voltage hyperpolarization induced cell death, a prediction is that chemically removing voltage before the onset of transients would protect cells, even if the cells are maintained in the presence of

aminoglycoside. If cells were treated with aminoglycoside, followed by CCCP addition, there would be an increase in the number of surviving cells compared to the removal of antibiotic, even if those cells were maintained in the antibiotic for a longer period of time. To test this prediction, *E. coli* were treated with 10 µg/mL gentamicin and CFUs were counted at 60 min. At that time, CCCP was added to the medium, and cells were incubated for another 60 min with aminoglycoside and CCCP. After 2 hr, CFUs were counted again, and there was a 22x increase in CFUs as compared to the 1 hr time point (*Figure 4E*). This data shows that the conditions for cell death had been established at 1 hr and that cells then plated onto LB would still die. However, cells treated with CCCP at 1 hr avoided the hyperpolarization-induced calcium transients and had a correspondingly higher survival rate.

## ATP dysregulation precedes voltage-induced bactericidal killing

Published evidence suggests that metabolic dysfunction correlates with translation inhibitor efficacy (*Levin et al., 2017*; *Allison et al., 2011*; *Lopatkin et al., 2019*). This was hypothesized to be associated with bacterial energetic investment in protein production (*Nieß et al., 2019*). Furthermore, a reduction in ribosome concentration has been annotated as a means to protect persister cells (*Cho et al., 2015*). We reasoned that the sudden change in energetic demand from the loss of a large fraction of 70S translating ribosomes could free up ATP and GTP to be used in other processes. To connect this shift in energetics to aminoglycoside-induced voltage dysregulation, we considered how *E. coli* generate a membrane voltage in aerobic environments. In the presence of glucose, *E. coli* use glycolysis to power the NADH dehydrogenase assembly (Complex I) and induce a proton motive force (PMF). The F1Fo-ATPase then depletes the PMF to generate ATP. However, the F1Fo-ATPase can be run in reverse, using ATP hydrolysis to generate a membrane voltage, which occurs in anaerobic conditions to power flagellar rotation (*Yasuda et al., 1998*). We hypothesized that aminoglycosides increased cellular ATP flux through non-ribosomal sinks, leading to hyperpolarization via the combined activity of the NADH dehydrogenase and a reversed F1Fo-ATPase.

We initially measured ATP concentration in *E. coli* using a ratiometric fluorescent ATP sensor, mRuby-iATPSnFR1.0 (*Lobas et al., 2019*). Gentamicin treatment increased the 488/561 nm fluorescence ratio by 50% within 2 hr of treatment (*Figure 5A*). Cells at low pH or in the presence of CCCP also showed ATP increases expected from ribosome dissociation (*Figure 4—figure supplement 1A, B*). Other non-aminoglycoside translation inhibitors which exhibit bacteriostatic activity also showed increasing ATP (*Figure 5—figure supplement 1C*). Consistent with our observation that recovered cells did not exhibit calcium transients, cells that recovered after 4 hr of kanamycin treatment had lower ATP compared to arrested cells (*Figure 5B*). We attempted to quantify the absolute change in ATP concentration in populations of cells, as our single-cell data indicated that ATP levels were increased when cells were treated with aminoglycosides. Using a luminescence-based assay, we determined that steady state levels of ATP in gentamicin-treated *E. coli* were significantly lower than untreated controls (*Figure 5—figure supplement 1D*) in the first half hour of treatment, which was inconsistent with our iATPSnFR single-cell data. This data is, however, consistent with an increased ATP flux through consumers other than ribosomes, such as the F1Fo-ATPase. We suspected that the genetically encoded ATP sensor can act as a buffer absorbing some of this ATP flux from a loss of translation, while the luminescence-based assay measures absolute values after the cells are permeabilized. This interpretation is consistent with recent results which show an increase in an alarmone with an ATP precursor after aminoglycoside treatment (*Ji et al., 2019*). Collectively, these data suggest that there may be a change in metabolic flux in the system and are consistent with prior observations of aminoglycoside-treated cells, which were found to leak NTPs (*Davis, 1987*) and increase respiration (*Lobritz et al., 2015*). This change in ATP flux is consistent with a number of other observations in the field correlating metabolism with translation inhibitor efficacy (*Levin et al., 2017*; *Allison et al., 2011*; *Lopatkin et al., 2019*; *Greulich et al., 2015*).

If aminoglycosides-induced ATP hydrolysis and hyperpolarization via the F1Fo-ATPase, then pump component knockouts should reduce calcium transients, show increased CFUs compared to WT, yet also show increased ATP due to the absence of hydrolysis. Knockouts from the proton conducting Fo domain (atpB, atpE, atpF) as well as atpG had increased CFUs and reduced calcium transients compared to WT (*Figure 5C,D* top), and all tested ATPase knockouts showed gentamicin-induced ATP accumulation (*Figure 5—figure supplement 1E*). Interestingly knockouts of atpC (ε-

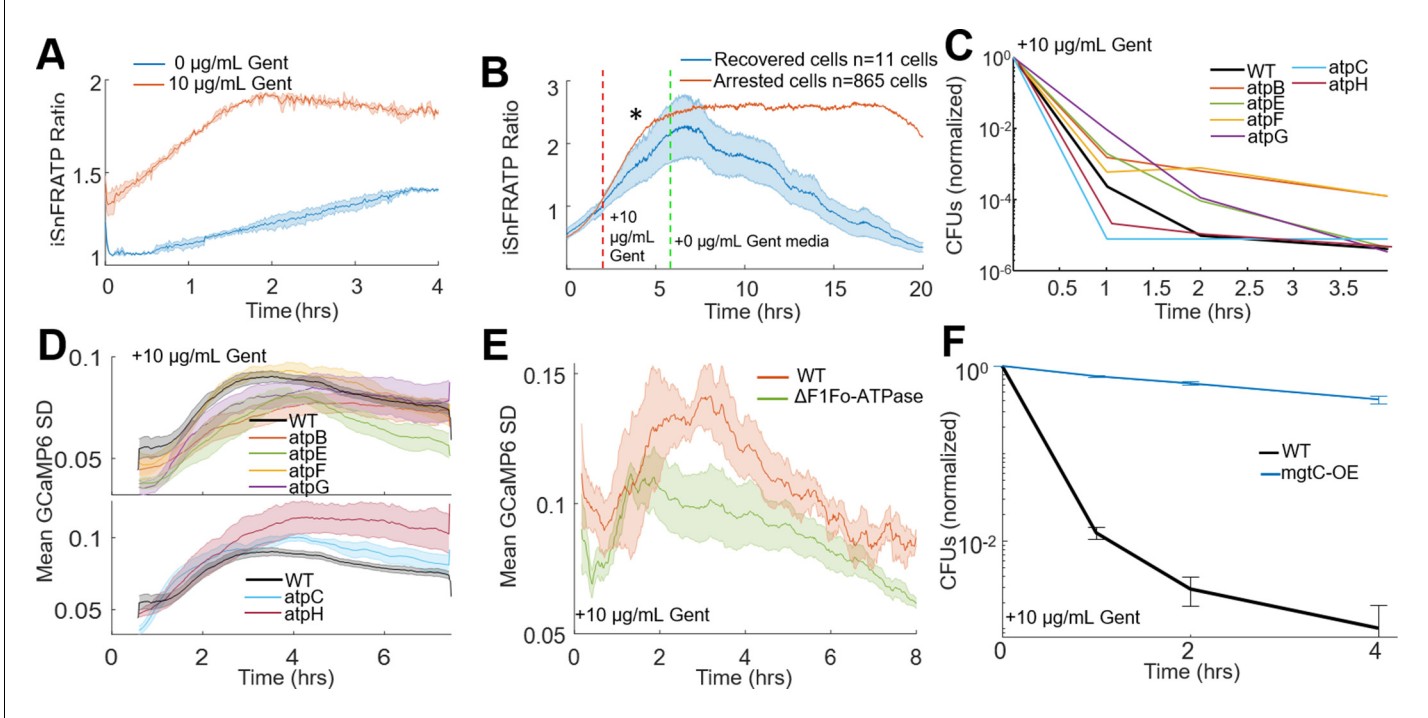

**Figure 5.** ATP dysregulation precedes voltage-induced bactericidal killing. (A) iATPSnFR ratios from *E. coli* treated with vehicle (blue) or 10 µg/mL gentamicin (orange). The ratio of iATPSnFR (488 nm) to mRuby (561 nm) indicates ATP concentration. Each trace averages two biological replicates. (B–F) Cells treated with 10 µg/mL gentamicin. (B) iATPSnFR ratios from gentamicin-treated cells that do (blue) or do not (orange) regrow. The star represents a significance of < 0.05 tested at 2 hr after treatment using a student t-test with unequal variance. (C) Normalized CFUs of gentamicin-treated knockouts of components of the F1Fo-ATPase compared to WT. Each data point is in biological triplicate. (D) Mean moving GCaMP6f SD for gentamicin-treated F1Fo-ATPase component knockouts compared to WT. Each curve averages four biological replicates. (E) Mean moving GCaMP6f SD for gentamicin-treated *E. coli* strain DK8, missing all components of the F1Fo-ATPase compared to WT. Each curve averages four biological replicates. (F) Normalized CFUs of gentamicin-treated mgtC expressing *E. coli* compared to WT.

The online version of this article includes the following figure supplement(s) for figure 5:

**Figure supplement 1.** ATP and membrane potential measurements are consistent with ATP dysregulation.

**Figure supplement 2.** Basal membrane potential measured in strains tested that protect against aminoglycosides does not explain the protective effect.

**Figure supplement 3.** Proposed model of aminoglycoside-induced cell death in *E. coli*.

subunit), which has increased gentamicin sensitivity (*Brynildsen et al., 2013*), and atpH (δ-subunit) both decreased the time to calcium transient onset and reduced CFUs faster than WT (*Figure 5C,D* bottom). AtpC biases the motor in the direction of ATP production (*Guo et al., 2019*), while atpH acts as a filter for proton conduction through the Fo domain (*Engelbrecht and Junge, 1990*), thus the knockouts of these proteins would improve proton conduction through the F1Fo-ATPase thereby increasing membrane potential that can be generated by this pump, which are consistent with knockouts showing more rapid cell death. Furthermore, gentamicin-treated Fo domain knockouts reduced hyperpolarization while, as their function predicts, atpC and atpH increased hyperpolarization relative to WT (*Figure 5—figure supplement 1F*). Completely eliminating the F1Fo-ATPase (Δ*unc* operon – strain DK8) (*Klionsky et al., 1984*) also showed reduced calcium transients as compared to a strain with intact F1Fo-ATPase activity (*Figure 5E*). Finally, expression of a virulence factor from *Salmonella*, mgtC, eliminated the bactericidal activity of aminoglycosides in *E. coli* (*Figure 5F*). MgtC is an inhibitor of the F1Fo-ATPase (*Lee et al., 2013*), and aids in *Salmonella* infection and survival at low magnesium (*Blanc-Potard and Groisman, 1997*; *Pontes et al., 2016*). To confirm the protective effects of the mgtC and strain DK8 were not due to a depolarized membrane potential, we measured basal membrane potential with TMRM and observed both were significantly more polarized as compared to WT (*Table 1*, *Figure 5—figure supplement 2*). Based on our model and previous data, hyperpolarization enhanced aminoglycoside killing in the absence of other

**Table 1.** Measurements of basal membrane voltage for protective strains

| Strain + treatment | Fluorescent dye | Mean emission (AU) | SD of emission (AU) | p-value (T-test) | Voltage estimates (mV) | Voltage estimate variance |
|---|---|---|---|---|---|---|
| BW+CCCP | TMRM | 399.86 | 104.09 | 0.02 | −133.66 | −34.79 |
| BW+None | TMRM | 716.67 | 47.76 | 1 | −150 | −10 |
| DK8+None | TMRM | 1123.41 | 218.67 | 0.08 | −162.59 | −31.65 |
| mgtC118+None | TMRM | 1062.52 | 202.43 | 0.09 | −161.03 | −30.68 |
| BW+CCCP | TMRM | 380.74 | 23.37 | <0.01 | −130.17 | −7.99 |
| BW+pH6.0 | TMRM | 535.72 | 24.2 | <0.01 | −139.73 | −6.31 |
| BW+pH7.5 | TMRM | 773.04 | 18.5 | 1 | −150 | −3.59 |
| DK8+pH7.5 | TMRM | 1674.41 | 8.67 | <0.01 | −171.64 | −0.89 |
| BW+CCCP | DiOC6 | 17.43 | 2.55 | 0.01 | −57.69 | −8.45 |
| BW+None | DiOC6 | 471.07 | 62.41 | 1 | −150 | −19.87 |
| DK8+None | DiOC6 | 724.67 | 162.21 | 0.1 | −162.06 | −36.27 |

protective effects, yet both of these strains show protected phenotypes, indicating that the loss or inhibition of the F-ATPase protected these strains relative to wildtype populations. These data were all consistent with aminoglycosides inducing membrane hyperpolarization from ATP hydrolysis via the F1Fo-ATPase, ultimately leading to cell death (*Figure 5—figure supplement 3*).

## Discussion

Aminoglycosides are well established to bind and exert pleiotropic effects on ribosomes (*Taber et al., 1987*; *Borovinskaya et al., 2007*; *Mehta and Champney, 2002*), and numerous reports highlighted the importance of maintaining a membrane potential in aminoglycoside activity. This evidence included voltage-dependent aminoglycoside uptake (*Leviton et al., 1995*) and cell death correlated with the citric acid cycle and carbon source (*Allison et al., 2011*; *Su et al., 2018*; *Meylan et al., 2017*). Metabolic changes can likewise induce changes in membrane voltage and the overall proton motive force. The relationship between metabolism, proton motive force, and membrane potential has been typically seen as being requisite to the *uptake* of aminoglycosides, which was synonymous with cell death (*Taber et al., 1987*; *Nichols and Young, 1985*). Our work has shown that membrane voltage is not essential for drug uptake, but rather the voltage is required to initiate the bactericidal mechanism after ribosome dissociation. Although we show a correlation between the ionic imbalance (calcium and pH transients) and cell death, we did not definitively prove they cause cell death, but rather they provide a convenient metric for cell death at the single-cell level. The GCaMP signal in our hands is certainly more accurate than PI uptake. Our data also does not preclude a mechanism of voltage enhanced aminoglycoside uptake (*Ezraty et al., 2013*; *Taber et al., 1987*). Rather our work suggests that the uptake of aminoglycosides in the absence of a membrane potential is sufficient to create intracellular conditions, including ribosome dissociation, metabolic dysfunction, and pore formation, that allow the presence of a membrane potential to exert bactericidal effects. Our data is also consistent with other translation inhibitors hyperpolarizing membrane potential correlated with subsequent cell death (*Lee et al., 2019*). We provide evidence that one mechanism by which this hyperpolarization can occur is through F1Fo-ATPase activity. We observed enhanced aminoglycoside killing in the strain atpC::kanR, which is missing the F1Fo-ATPase ε-subunit that typically biases the rotor in the direction of ATP synthesis. This observation suggests that F1Fo-ATPases with a higher likelihood of ATP hydrolysis enhance aminoglycoside killing, which would stem from the already ribosome-related dysregulation of metabolism. We observed similar enhanced aminoglycoside killing in the strain atpH::kanR that encodes the δ-subunit of the F1Fo-ATPase, which is able to block proton conduction (*Engelbrecht and Junge, 1990*) and ATP hydrolysis (*Xiao and Penefsky, 1994*). Together, these data suggest that the difference between bactericidal activity of aminoglycosides compared to the bacteriostatic activity of other translation inhibitors may be the lack of the mistranslated membrane proteins causing pore

formation. We hypothesize this mechanism kills bacteria by eliminating ion homeostasis in the presence of a membrane potential and pores that can leak ions. However, we currently lack tools to be able to induce the calcium transients in the absence of aminoglycosides, although perhaps channel rhodopsins will be able to mimic these effects.

One fascinating facet that remains to be explored is the period after aminoglycoside treatment that cells cannot divide but remain metabolically active for at least 2 days. If these arrested cells can still export quorum-sensing molecules, they could send paracrine signals to untreated cells, and influence their behavior. This observation became clear by using sensitive genetically encoded fluorescent proteins, and these tools open up a new avenue to study the long-term effects of antibiotic treatment on cells and mixed cultures. Another curious corollary is the observation that protonophores enhance aminoglycoside killing in *Pseudomonas* biofilms (*Maiden et al., 2018*; *Maiden et al., 2019*), which stands in opposition to our observation that protonophores protect planktonic *E. coli*. The differences driven by these species specific and context-dependent observations will hopefully add to a more complete picture of aminoglycoside activity in multiple bacterial species.

The model of aminoglycoside-induced death proposed from this work is consistent with evidence from other groups previous work, requires the presence of membrane pores and membrane potential to drive aminoglycoside bactericidal activity (*Figure 5—figure supplement 3*). Aminoglycosides enter the cell through an unknown mechanism, possibly through channels such as mscL (*Wray et al., 2016*), which occurs long before a loss of membrane integrity. Once aminoglycosides enter the cell they bind ribosomes, disrupt a majority of translating 70S particles and cause mistranslation of protein (*Kohanski et al., 2008*; *Dubin and Davis, 1961*). As soon as ribosome disruption occurs, respiration (*Lobritz et al., 2015*) and metabolism (*Levin et al., 2017*) go through a substantial shift in flux. This disruption of metabolism enables non-canonical generators of membrane potential, such as the F1Fo-ATPase to drive changes in membrane potential. Why voltage is so toxic in the presence of the mistranslated membrane proteins remains to be explored; however, this shift in understanding the role voltage plays in aminoglycoside lethality will hopefully provide a necessary rethinking of how these antibiotics function so much more effectively than other translation inhibitors. The difference between these mechanisms of bactericide and stasis could lead to novel antibiotics that impinge on the aminoglycoside mechanism of action.

## Materials and methods

**Key resources table**

| Reagent type (species) or resource | Designation | Source or reference | Identifiers | Additional information |
|---|---|---|---|---|
| Strain, strain background (*Escherichia coli*) | *E. coli* K-12 BW25113 | Yale Coli Genetic Stock Center | CGSC#: 7636 | |
| Strain, strain background (*Escherichia coli*) | BW25113 ΔnuoA | Dharmacon Keio | OEC4987-213603796 | |
| Strain, strain background (*Escherichia coli*) | BW25113 ΔnuoB | Dharmacon Keio | OEC4987-213603795 | |
| Strain, strain background (*Escherichia coli*) | BW25113 ΔnuoH | Dharmacon Keio | OEC4987-213603791 | |
| Strain, strain background (*Escherichia coli*) | BW25113 ΔnuoI | Dharmacon Keio | OEC4987-213603790 | |
| Strain, strain background (*Escherichia coli*) | BW25113 ΔatpA | Dharmacon Keio | OEC4987-213606163 | |

*Continued on next page*

*Continued*

| Reagent type (species) or resource | Designation | Source or reference | Identifiers | Additional information |
|---|---|---|---|---|
| Strain, strain background (*Escherichia coli*) | BW25113 ΔatpB | Dharmacon Keio | OEC4987-213606167 | |
| Strain, strain background (*Escherichia coli*) | BW25113 ΔatpC | Dharmacon Keio | OEC4987-213605824 | |
| Strain, strain background (*Escherichia coli*) | BW25113 ΔatpD | Dharmacon Keio | OEC4987-213605825 | |
| Strain, strain background (*Escherichia coli*) | BW25113 ΔatpE | Dharmacon Keio | OEC4987-213606166 | |
| Strain, strain background (*Escherichia coli*) | BW25113 ΔatpF | Dharmacon Keio | OEC4987-213606165 | |
| Strain, strain background (*Escherichia coli*) | BW25113 ΔatpG | Dharmacon Keio | OEC4987-213607977 | |
| Strain, strain background (*Escherichia coli*) | BW25113 ΔatpH | Dharmacon Keio | OEC4987-213606164 | |
| Strain, strain background (*Escherichia coli*) | *E. coli* DK8 1100Δ(uncB-uncC)ilv::TnlO | Rubinstein lab (created in *Guo et al., 2019*) | DK8 | |
| Strain, strain background (*Escherichia coli*) | BL21(DE3) | Sigma-Aldrich | CMC0016 | Electrocompetent cells |
| Transfected construct (bacterial) | Plasmid: pKL09-GCaMP6f-mScarlet bb118 | This paper | Addgene #pending | GCaMP6-mScarlet |
| Transfected construct (bacterial) | Plasmid: pKL10-mRuby-iATPSnFR1.0 bb118 | This paper | Addgene # pending | mRuby-ATPSnfr |
| Transfected construct (bacterial) | Plasmid: pKL12-GCaMP pHuji bb118 | This paper | Addgene # pending | GCaMP6-pHuji |
| Transfected construct (bacterial) | Plasmid: pKL13-MgtC bb118 | This paper | Addgene # pending | S. typhimurium mgtC expression plasmid |
| Transfected construct (bacterial) | Plasmid: pKL11-GCaMP6f bb100 | This paper | Addgene # pending | GCaAMP6 |
| Chemical compound, drug | Glucose | Sigma | G7528-1KG | |
| Chemical compound, drug | M9 salts | Sigma | M6030 | |
| Chemical compound, drug | MEM amino acid | Gibco | 11130–051 | |
| Chemical compound, drug | Glutamate | Sigma | G1251 | |
| Chemical compound, drug | Hydrochloric acid | Sigma | 320331 | |
| Chemical compound, drug | Low melt agarose | VWR | 97064–134 | |
| Chemical compound, drug | Sodium hydroxide | Sigma | 795429 | |
| Chemical compound, drug | Kanamycin sulfate | Sigma | 60615–5G | |

*Continued on next page*

*Continued*

| Reagent type (species) or resource | Designation | Source or reference | Identifiers | Additional information |
|---|---|---|---|---|
| Chemical compound, drug | Gentamicin sulfate | Sigma | 345814 | |
| Chemical compound, drug | Apramycin | Sigma | A2024-1G | |
| Chemical compound, drug | Streptomycin sulfate | Sigma | S9137 | |
| Chemical compound, drug | Tobramycin | Sigma | 614005 | |
| Chemical compound, drug | Trimethoprim | Sigma | T7883 | |
| Chemical compound, drug | Cyclohexamide | Sigma | C7698 | |
| Chemical compound, drug | Chloramphenicol | Sigma | C1919 | |
| Chemical compound, drug | Erythromycin | Sigma | E5389 | |
| Chemical compound, drug | Potassium chloride | Sigma | P9333 | |
| Chemical compound, drug | Magnesium chloride | Sigma | 63068 | |
| Chemical compound, drug | CCCP | Sigma | C2759 | |
| Chemical compound, drug | Oxyrase for broth | Sigma | SAE0013 | |
| Chemical compound, drug | Texas Red-X, Succinimidyl Ester | Thermo-Fischer | T20175 | |
| Chemical compound, drug | N,N-Dimethyl formamide | Sigma | 227056 | |
| Chemical compound, drug | Glutathione | Sigma | G4251 | |
| Chemical compound, drug | Ascorbic acid | Sigma | A7506 | |
| Chemical compound, drug | Propidium iodide | Life-tech | P3566 | |
| Chemical compound, drug | Tetramethylrhodamine, methyl ester | Molecular Probes | T668 | |
| Chemical compound, drug | DL-Dithiothreitol | Sigma | D9779 | |
| Chemical compound, drug | Lysozyme from chicken egg white | Sigma | 62971–10 G-F | |
| Chemical compound, drug | Magnesium chloride hexahydrate | Sigma | 63068–250G | |
| Chemical compound, drug | Sucrose | Sigma | 84097–1 KG | |
| Chemical compound, drug | Water, Sterile. WFI Quality | Sigma | 4.86505.1000 | |
| Chemical compound, drug | Sodium deoxycholate | Sigma | 30970–25G | |
| Chemical compound, drug | Ammonium chloride | Sigma | 09718–250G | |
| Chemical compound, drug | $DiOC_6(3)$ | Sigma | 318426–250 MG | |

*Continued on next page*

*Continued*

| Reagent type (species) or resource | Designation | Source or reference | Identifiers | Additional information |
|---|---|---|---|---|
| Software, algorithm | MATLAB | https://www.mathworks.com/products/matlab.html | RRID:SCR_001622 | |
| Software, algorithm | NIS Elements | https://www.microscope.healthcare.nikon.com/products/software/nis-elements | RRID:SCR_002776 | |

## Lead contact and materials availability

Plasmids generated in this study are available on Addgene. Knockout strains from the Keio collection are available through Dharmacon due to an MTA. Other reagents are available upon request, and will be fulfilled by the Lead Contact, Joel Kralj (joel.kralj@colorado.edu).

## Experimental model and subject details

### *E. coli* strains

*E. coli* strain BW25113 was acquired from the Yale Coli Genetic Stock Center and was used as the control, except experiments where specifically noted. Knockout strains were acquired from the Keio collection purchased from Dharmacon (#OEC4988). *E. coli* strain DK8 1100Δ(uncB-uncC)ilv::TnlO, which is deficient of the F-ATPase was a generous gift from the Rubinstein lab.

### Cell growth

Strains were grown in LB with antibiotics dependent on growth conditions. For GCaMPmScarlet expressing cells, clones transformed with the plasmid were grown overnight with carbenicillin (100 μg/mL). Carbenicillin was used for overnight cultures to maintain the plasmid but was not present for any experiments. For knockout strains from the keio collection kanamycin (50 μg/mL) was also added to any overnight cultures. Strain DK8 was grown overnight in the presence of tetracycline (30 μg/mL). Glycerol stocks were streaked onto plates bearing the appropriate antibiotics, and individual colonies were picked and grown in 5 mL culture tubes, or in 24-well plates, or in 50 mL Erlenmeyer flasks. All cells were grown overnight at 37°C with shaking between 150 and 200 rpm with the appropriate antibiotic if required for plasmid or strain selection. Knockouts from the Keio collection were plated on LB plates with kanamycin and carbenicillin to ensure maintenance of the knockout cassette, but overnight liquid cultures that were to be used for imaging were grown only in the presence of carbenicillin to avoid any potential effects of protein translation inhibition on sensor expression.

## Method details

### Plasmids

Expression of GCaMP6f-mScarlet was carried out with a constitutive promoter (118, iGem biobrick) assembled in an ampicillin-resistant plasmid similar to earlier work (*Bruni et al., 2017*). The mScarlet amino acid sequence was taken from the original publication (*Bindels et al., 2017*) and purchased as a gBlock (IDT). The plasmid was double digested with Pme1/Nco1 and assembled using Gibson assembly. The mRuby-iATPSnFR1.0 construct was created by obtaining the amino acid sequence directly from the publication (*Lobas et al., 2019*) and codon optimizing it in a single gBlock ordered from IDT, then Gibson cloned into the same constitutive promotor backbone as GCaMP6f-mScarlet. Expression of these constructs was carried out in the 118 plasmid. Expression of GCaMP6f alone was carried out using a similar constitutive promoter (100, iGem biobrick) in the same backbone. The mgtC over expression plasmid was created by obtaining the amino acid sequence directly from salmonella on a gBlock, and Gibson cloned into the 118 biobrick backbone used above. GCaMP6f tethered to pHuji was purchased on a gBlock and Gibson cloned into the same constitutive biobrick 118 promoter used previously. All novel plasmids and sequences have been deposited on Addgene. All plasmids were transformed into their respective genetic background strain using Transfer Storage Solution transformation protocol.

## Imaging media and fluorescent dyes

Unless otherwise noted, all imaging experiments were conducted in PMM at pH 7.5. The PMM recipe used is: 1x M9 salts (Sigma), 0.2% glucose (Sigma), 0.2 mM $MgSO_4$, 10 µM $CaCl_2$, 1x MEM amino acids (Gibco). Experiments were conducted at pH 7.5 unless otherwise noted in the text, and NaOH or HCl was used to change the pH to the final value. Given the critical importance of pH in aminoglycoside response, all PMM media with additional chemicals was pH adjusted to 7.5 before imaging. At more basic pH, and higher concentrations of Mg, precipitate forms in this media over time. For oxygen free microscopy experiments, Oxyrase for Broth was added to the media pads during the pre-imaging incubation time to 10% v:v, then sealed to have oxygen removed.

Propidium iodide (Life Tech) was dissolved in water in a stock concentration, and added to a final concentration of 3 µg/mL. PI was imaged with a 561 nm laser in a flow experiment, and was added at the same time as 30 µg/mL kanamycin.

Gentamicin Texas Red (GTTR) was synthesized using a previously described protocol (*Saito et al., 1986*). Texas Red-succinimidyl ester (Invitrogen) was dissolved in anhydrous *N,N*-dimethylformamide on ice to final concentration of 20 mg/ml. Gentamicin was dissolved in 100 mM $K_2CO_3$, pH 8.5, to a final concentration of 10 mg/ml. On ice, 10 µL of Texas Red was slowly added to 350 µL gentamicin solution to allow a conjugation reaction. The gentamicin-Texas Red product from this reaction was used for the imaging experiments. Gentamicin uptake was measured by incubating gentamicin-Texas Red (final concentration of 10 µg/ml) simultaneously with GCaMP6f in a flow experiment.

TMRM (ThermoFischer) was dissolved in DMSO in a 1 mM stock solution, and diluted in PMM to 8 µM, then added to a final concentration of 200 nM to cell suspensions. TMRM was measured as described below in flow cytometry.

$DiOC_6(3)$ was dissolved in Ethanol to a 10 mM stock solution, and diluted in PMM to 1 mM, then added to a final concentration of 1 µM to cell suspensions. $DiOC_6(3)$ was measured as described below in flow cytometry.

## Preparing cells for imaging

All imaging of cells took place under agarose pads which were composed of PMM at the appropriate pH and 2% low melt agarose.

For experiments using flow, the agarose was melted in PMM buffer and cast between 2 pieces of glass covering a silicone mold. The silicone was 3/16' as the final thickness, and was cut by hand. The pads were diced into small squares using an exacto knife to fit into the flow chambers (~2 mm x 2 mm). Cells from an overnight culture were placed directly on to the agarose pad (1.0 µL) and left for ~5 min. The agarose pads were then placed with the cells down onto a 24 mm x 50 mm glass coverslip (thickness 1.5) with a silicone flow chamber. The apparatus was then sealed with a custom glass slide with holes drilled to enable flow.

Experiments involving drug titrations or knockouts were prepared onto 96-well glass bottom dishes (Brooks Automation, MGB096-1-2-LG-L). A custom 96-well mold was created and 3D printed using a commercial service. The mold was designed to hold a volume of 200 µL per well (Shapeways), with a separate piece designed to press the agarose pads into the coverslip in 8-, 12-, or 96-well format. The 3D printed pieces are available at the Kralj Lab store on Shapeways (https://www.shapeways.com/shops/kraljlab) and the. stl files are available to researchers upon request. The bottom of the agarose mold was sealed with a 4' x 6' piece of glass (McMaster Carr), and liquid agarose was added to the desired wells. A second piece of glass (3' x 5') was used to seal both sides, and the agarose was left to cast for >1 hr. The glass piece was then removed, and cells were added to each pad individually (2 µL) and left for 10 min for the liquid to absorb into the agarose. The cells were then pressed out into the 96-well plate using the custom 3D printed press. For all experiments, cells were left in the pad for ~1 hr before imaging. Any chemical treatments were then added to the top of the pad. A 5 µL drop of a solution at 40x final concentration was added on to the pad and left to diffuse throughout. In house measurements with a small fluorescent dye showed compounds diffuse to the glass in ~5 min.

## Imaging

Flow experiments were conducted using a Nikon TiE base with perfect focus, running Elements software, with a custom laser illumination with high angle illumination. A 488 nm (Obis 150 LX,

Coherent) or 561 nm (Obis 50 LS) were combined, expanded, and focused onto the back aperture to create a widefield illumination. A mirror located 1 f away from the widefield lens was used to control the illumination angle. Imaging took place with a 100x NA 1.45 objective with intensities (at the sample) of 130 mW/cm$^2$ 488 nM and 1050 mW/cm$^2$ 561 nm light. A quad band emission filter (Semrock) was used for reflecting the illumination light, and no emission filter was needed. The light was imaged onto an Andor EMCCD (iXon 888 Ultra) using an exposure time of 200 ms. Images were acquired sequentially (561 nm, then 488 nm) once per minute over the entire experiment (6–48 hr). These illuminations showed no evidence of phototoxicity compared to unilluminated cells as measured by growth rate.

Flow was controlled with two identical syringe pumps (Harvard Apparatus). Flow rates were set to 20 µL/minute which was sufficient to fully exchange the medium in the chamber within 2.5 min. Each syringe pump was loaded with the appropriate medium and was programmed to turn on or off at the desired time. A typical experiment involved 2 hr of PMM alone, followed by switching to PMM +Kan using the second pump. Tubing from multiple syringes was connected with a T-connector with a dead volume of ~20 µL. At all times during flow cell experiments, the specified media was flowed through the chamber.

Imaging 96-well glass bottom plates took place using a Nikon Ti2 inverted microscope running the Elements software package. Fluorescent excitation was achieved with a Spectra-X LED source (Lumencor). A 40x, NA 0.95 air objective was used to both illuminate and image the cells onto 2-Flash 4 v2 sCMOS cameras (Hamamatsu) using a custom splitter to image two colors simultaneously (Thorlabs). Illumination was achieved by simultaneous excitation with 470/26 and 554/20 band pass LED illumination for a 200 ms exposure. Measured light intensities at the sample were 330 mW/cm$^2$ (470 nm) and 2050 mW/cm$^2$ (554 nm). Typical sampling rates were one frame per minute, unless noted in the text.

## CFU measurements

CFUs were measured by plating-treated cells onto LB-agarose without antibiotic and counting growing colonies. CFU measurements were conducted trying to mimic the experiments performed via microscopy. Briefly, cells were grown overnight in LB and diluted 1:20 in 5 mL PMM. These cultures were grown at room temperature and shaking for 2 hr (t = 0) followed by the addition of antibiotic. At each time point, the culture was removed from the shaker, and 100 µL was removed. A 10x series dilution was then conducted by removing 20 µL and adding to 180 µL LB alone in a 96-well plate. The 10-fold dilution was performed seven times, leading to the original concentration to a dilution of 10$^7$. From each of the 10x dilution series, 3 µL was plated onto an LB agar pad and left to dry (one colony = 333 cells/mL, lower end of our dynamic range). After an entire experiment (typically 5 hr), the agar was placed into an incubator and grown overnight. Colonies were then manually counted the next morning.

## Cell cytometry

A 5 ml of PMM media was seeded with 50 µL of overnight BW25113 cells, or the respective knockout strain tested. When the cells reached ~0.4 OD, 100 µg/ml kanamycin, 10 µg/ml gentamicin, or PMM alone was added. After 30 min of antibiotic or mock treatment, TMRM or DiOC$_6$(3) was added to the suspension at a final concentration of 0.2 mM or 1 µM, respectively. Two hours later, 1 mL of cell suspension was transferred to a 1 mL Falcon polystyrene round-bottom tube. Cells were quantified for their TMRM incorporation by counting 100,000 events per condition using a BDFACSCellesta Flow Cytometer with the following Voltage settings: FSC at 700, SSC at 350, with 561 nm laser D585/15 at 500, C610/20 at 500 and B670/30 at 481. Emission for each event was collected at the 585/15 nm wavelengths. Cells were quantified for their DiOC$_6$(3) incorporation by counting 100,000 events per condition using a BDFACSCellesta Flow Cytometer with the following Voltage settings: FSC at 700, SSC at 350, with 488 nm laser B 530/30 at 350. Emission for each event was collected at the 530/30 nm wavelengths.

## Bactiter glo ATP analyses

ATP per optical density unit was quantified using Promega's BacTiter-Glo kit coupled with a BioTek Synergy plate reader. BacTiter-Glo reagents and standards were prepared as described in the

manual. Briefly, exponentially growing cultures of *E. coli* were treated according to the experimental parameters for the times indicated. When the time of treatment was reached 100 μL of culture, blank, or standard, was added to a black walled clear bottom 96-well plate. This was done in technical triplicate for each condition, blank, or standard, which had at least three biological replicates. Once the plate was prepared 100 μL of BacTiter-Glo Reagent was added to each well, and shaken in an orbital shaker for 1 min at room temperature, then left on the benchtop for 5 min. Luminescence was recorded using the BioTek Synergy plate reader, set to auto scaling and 1 s integration time. Simultaneously with BacTiter-Glo plate preparation, an optical density plate was created with the same cultures, and the absorbance of the culture was read on the same plate reader at 600 nm. ATP per OD unit was calculated by the average of the three biological replicates (which were averaged from the technical replicates), which were then divided by the obtained OD values.

## Polysome analyses

Sucrose gradients were prepared in Beckman Coulter Ultra-Clear Tubes (14 × 89 mm) Reorder No. 344059. Media recipes and protocol is from *Qin and Fredrick, 2013*. Roughly 6 ml of 10% sucrose was layered on the bottom of the tube, then a large needle was used to add 40% sucrose below the 10% layer up to a 6 ml marker on the outside of the tube. If a clear meniscus between the two layers was not visible the tube was discarded. Tubes were placed in a MagnaBase tube holder (sku B105-914A-I/R), and short caps were placed on top to eliminate all air from the tube. The tube holder was then placed on the gradient maker. A 10–40% gradient was then established using a BIOCOMP Gradient Station ip gradient maker with the following settings: Short cap, Sucrose, 10–40%wv, 81°, 1:48 min:sec. Caps were then removed, and gradient tubes were stored no longer than 1 hr at 4 °C until lysate supernatant was prepared.

Ribosomes and ribosomal subunits were characterized using a slightly adapted protocol, due to differences in available equipment, from *Qin and Fredrick, 2013*. Briefly 50 mL cultures were grown to ~0.35–0.45 OD. Antibiotic, or a mock treatment was then added, and these cultures were allowed to grow for another 1 (LB, *Figure 1B*) or 1.5 (PMM, *Figure 1C*). Aminoglycosides enter cells and induce ribosomal dissociation in the abscence of membrane voltage.; *Figure 1—figure supplement 1*. Aminoglycosides enter cells and induce ribosomal dissociation in the absence of membrane voltage.B1-S1BF1F1-S1,) hr. Optical Density was taken at time of collection, when 37.5 ml of culture was then transferred to Nalgene Oak Ridge Centrifuge Tubes (Cat. 3119–0050) on ice. Cells were then pelleted in a chilled Sorvall SA-600 rotor in a Sorvall RC 5C Plus Centrifuge at 10,000 rpm for 5 min at 4 °C. Culture media was decanted and aspirated. Cell pellets were then resuspended in 500 μL lysis buffer (750 μL for anaerobic conditions), and flash frozen in liquid nitrogen. Frozen suspensions were thawed in a 5–10°C water bath, then flash frozen again, and either stored at −80 °C or thawed in the same manner and treated as follows. Lysis was completed by adding 15 μL of 10% sodium deoxycholate to freeze-fractured pellet resuspensions and mixed by inversion. Lysate was then separated by centrifugation at 4 °C at 10,000 rpm in a chilled Eppendorf FA45-30-11 rotor in an Eppendorf 5804R Centrifuge for 10 min at 4 °C. Lysate supernatant was collected in chilled microfuge tubes. Then 300 μL of the 10–40% gradient was removed from the top of the sucrose gradient columns and replaced with 300 μL of lysate supernatant.

Loaded gradient columns were placed in Beckman SW-41 swinging buckets and balanced to within 0.01 g of each other using the 10% sucrose solution. Loaded sucrose gradient buckets were then centrifuged using the SW-41 rotor in an LM-8 Ultracentrifuge in 4 °C at 35,000 rpm for 3 hr. Sucrose gradient columns were then removed, and fractions were then collected using the following series of machines. A BIOCOMP Gradient Station IP with settings Distance 80.00 mm, Speed 0.3 mm/s was tethered to a BIORAD Model 2110 Fraction Collector with the following settings: six drops/fraction. As fractions were collected the absorbance at 254 nm was collected from the fractions using a BIORAD Econo UV Monitor set to range 1.0 (AUFS) tethered to computer running the BIOCOMP Gradient Profiler 2.0 software. Data files for each gradient run were saved as. csv files and later analyzed in Matlab using custom scripts to integrate peaks with the trapz.m function.

Due to the nature of collection with these devices, often the beginning of the non-ribosomal RNA peaks was missed, capturing the absorbance as the non-ribosomal RNA ran through the detector midway through the peak. In all conditions tested, non-ribosomal RNA, 30S, 50S, and 70S peaks were detected. To simplify comparisons between conditions, polysomes beyond the 70S peak were

ignored in the (30S+50S)/70S ratio measurements. Note that because of the nature of these experiments, different total quantities cell lysate, and therefore of total RNA, are loaded into the sucrose gradients columns. Due to this reality, comparing the 254 nm absorbance quantities between samples is unreasonable; however, comparing the ratio of the ribosome peaks should be total-RNA agnostic.

## Quantification and statistical analysis

### Image processing

Data was stored as. ND2 files which contain the 16 bit images and the associated metadata. The Bio-Formats Matlab package was used to access data in the. ND2 format. All data analyses were performed using custom scripts in Matlab (available upon request).

Image processing followed the general scheme of (1) estimating the illumination profile for all experiments on a given day, (2) correcting the uneven illumination for each movie, (3) registering drift and jitter in XY, (4) subtracting an estimated background, (5) segmenting cells using a Hessian algorithm, (6) extracting time traces for individual cells, (7) processing each time trace for the onset and amplitude of calcium transients.

1. Estimating the illumination profile: For a given day, every movie was averaged across time, and opened using a morphological operator and blurred using a 2D Gaussian filter. Each of these experimental images were then averaged together to give an estimate of the uneven illumination. These images were smooth across the entire field of view, and varied by ~50% across the entire image.

2. Correcting uneven illumination. Each individual movie was then loaded into memory sequentially. Each frame of the movie was converted to a double, and then divided by the uneven illumination. This image was then multiplied by the average value of the movie and converted back into a uint16 to maintain consistent intensity values. Each frame was then reassembled into an illumination corrected movie.

3. Registering drift and jitter in XY: Each frame was aligned to the previous frame using a convolution of the 2D Fourier transform. Each sequential image was first estimated by applying the XY warping from the previous frame. Then, the 2DFT was taken for each image, and multiplied to the previous frame. The optimal updated XY position was then calculated and applied.

4. Subtracting the estimated background: The background was estimated for each frame individually using a morphological operator. A disk structured element with radius 9 μm was blurred with a Gaussian filter. This background estimation was then subtracted from the original image. To protect against potential negative values, the minimum of the entire movie was set to 50 counts.

5. Segmenting cells using a Hessian algorithm: To segment cells, first the foreground was estimated using Otsu's method from the background subtracted image. The Hessian was then calculated on the background subtracted image, and then elementwise multiplied to a logical image of the foreground. Otsu's method was again used on this modified Hessian image to identify individual cells. Hard limits were set to remove potential noise that did not fit given criteria for size or minimum intensity. We found that first increasing the size of the image using a spline interpolation gave superior segmentation results. Using this method, not all cells were identified within a microcolony, though we estimate that it can identify ~96% of the cells accurately.

6. Extracting time traces for individual cells: From a given identified cell, for each time point in the movie, we extracted the mean intensity using the Matlab command, regionprops. The mean intensity for both the GCaMP6f and the mScarlet were extracted using this method, or any other fluorophore the cells expressed.

7. Processing each time trace for the onset and amplitude of transients: For each time trace, the moving median over 45 min was divided to remove the slow baseline trends. A standard deviation was calculated from the timepoints before aminoglycoside addition, and a cell was defined as blinking if it had transients that lasted >10 min that were >7 x the pre-treatment standard deviation. From each normalized time trace, the moving standard deviation was also calculated using a 30 min sliding window. Within a given FOV, the entire population moving standard deviations was averaged, providing the average standard deviation trace shown in the figures.

During flow experiments, single frames were sometimes contaminated by bubbles that dramatically changed the contrast. To remove these features, we took the average of all extracted cells. If

the differential of any single frame was initially lower, then higher than 5x the standard deviation of the whole movie, a single frame was removed. This preprocessing removed spurious catastrophic blinks that appeared to occur in every cell at the same instant.

## Cytometry analysis

*E. coli* energize their membrane through a proton motive force (PMF) that powers their flagellar motors and several membrane pumps. The PMF is the amount of free energy gained by a proton moving from one side of the membrane to the other, and the energy can be gained either by changes in pH (proton gradient) or voltage (membrane potential). The Nernst equation sets an equivalence between changes in pH and voltage as:

$$PMF = \psi * -58mV(\Delta pH)$$

*E. coli* typically try to maintain a cytoplasmic pH around 7.5, so that a changing extracellular pH will induce a corresponding change in the PMF. For example, if the extracellular pH is at pH 7.5, then there is no pH difference, so all of the PMF will be carried in the voltage component, which would be accomplished by establishing ionic gradients using pumps and channels. On the other hand, if the extracellular pH is low, for example pH 6, then the PMF could have a value $-87$ mV without having to maintain any voltage component. The PMF could be carried entirely by the change in pH, which could drive the flagellar motors and other PMF dependent processes in the membrane.

Thus, by changing the environmental pH from 7.5 to 6, we can lower the membrane voltage by fact that the cell will utilize the pH component of PMF without the need to generate an external voltage from other ions.

Analysis of the cytometry data was achieved by fitting the data to a 1D Gaussian distribution and calculating the mean and 95% confidence interval for each of these fits for each strain tested. These values were then taken as a ratio of the gentamycin-treated cells relative to the vehicle-treated cells.

Assuming TMRM partitions according to Boltzmann's law:

$$\frac{C_{in}}{C_{out}} = e^{\frac{-q}{kT}V}$$

where $C_{in}$ and $C_{out}$ are the concentrations of the dye in and out of the cell, q is the ionic charge, k is the Boltzmann constant, and T is the temperature in kelvin. Comparing two different conditions ($V_{kan}$ and $V_{PMM}$), we can solve for the treated condition to yield:

$$V_{Kan} = V_{PMM} - 28mV * ln\left(\frac{C_{in,Kan}}{C_{in,PMM}}\right)$$

if we assume the concentration of dye out of the cell is the same in both conditions. Given a large reservoir relative to the cytoplasmic volume of the cells, this is a reasonable estimate. These same assumptions and calculations were applied to the values in *Table 1*, as well as for the dye DiOC$_6$(3).

## Significance testing

Significant differences across populations of individual cells were tested using the unpaired t-test with unequal variance. For cytometry experiments, we used the 95% confidence interval (CI) to a single Gaussian fit.

## Acknowledgements

Thanks to Annette Erbse and Keda Zhou for help with ribosomal profiling, Theresa Nahreini for help with cytometery, and Karolin Luger, Joe Falke, Tom Cech and the Biochemistry Shared Instruments Pool at the University of Colorado Boulder for equipment resources. We thank Stephanie Bueler and John Rubinstein for the DK8 strain and helpful discussion. We had helpful discussions with Corrie Detweiler, Amy Palmer, Ben Dodd, Stacey Scott, and Rose Luder. Special thanks to Thomas Yao for many helpful discussions. Funding: Searle Scholars Program and NIH New Innovator (1DP2GM123458) to JMK, T32 training grant (T32GM065103) and HHMI Gilliam Fellowship for Advanced Study to GNB Flow cytometer was acquired with an instrumentation grant (NIH S10OD021601).

## Additional information

### Funding

| Funder | Grant reference number | Author |
|---|---|---|
| Howard Hughes Medical Institute | Gilliam Fellowship for advanced study | Giancarlo Noe Bruni |
| Kinship Foundation | Searle Scholar Award | Joel Kralj |
| National Institute of General Medical Sciences | T32GM065103 | Giancarlo Noe Bruni |
| National Institute of General Medical Sciences | 1DP2GM123458 | Joel Kralj |

The funders had no role in study design, data collection and interpretation, or the decision to submit the work for publication.

### Author contributions

Giancarlo Noe Bruni, Software, Formal analysis, Investigation, Methodology, Writing - original draft, Writing - review and editing; Joel M Kralj, Conceptualization, Data curation, Software, Formal analysis, Supervision, Funding acquisition, Investigation, Methodology, Writing - original draft, Project administration, Writing - review and editing

### Author ORCIDs

Giancarlo Noe Bruni (iD) https://orcid.org/0000-0003-2850-4633

Joel M Kralj (iD) https://orcid.org/0000-0001-9370-2324

### Decision letter and Author response

Decision letter https://doi.org/10.7554/eLife.58706.sa1

Author response https://doi.org/10.7554/eLife.58706.sa2

## Additional files

### Supplementary files

• Transparent reporting form

### Data availability

All data have been submitted on Dryad.

The following dataset was generated:

| Author(s) | Year | Dataset title | Dataset URL | Database and Identifier |
|---|---|---|---|---|
| Bruni GN, Kralj JM | 2020 | *E. coli* aminoglycoside treatment | https://doi.org/10.5061/dryad.fxpnvx0pp | Dryad Digital Repository, 10.5061/dryad.fxpnvx0pp |

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
