## [Decision Letter]

**Acceptance summary:**

Although aminoglycosides are important, clinically relevant antibiotics; the precise mechanism by which they kill bacteria has remained unclear and somewhat controversial. This paper uses clever imaging approaches based on a voltage-sensitive reporter to provides evidence that their bactericidal activity stems from a hyperpolarization of the inner membrane of cells rather than affecting uptake, or at least rather than affecting only uptake.

**Decision letter after peer review:**

[Editors’ note: the authors submitted for reconsideration following the decision after peer review. What follows is the decision letter after the first round of review.]

Thank you for submitting your work entitled "Membrane voltage dysregulation, not uptake, underlies bactericidal activity of aminoglycosides" for consideration by *eLife*. Your article has been reviewed by three peer reviewers, one of whom is a member of our Board of Reviewing Editors, and the evaluation has been overseen by a Senior Editor. The reviewers have opted to remain anonymous.

Our decision has been reached after consultation between the reviewers. Based on these discussions and the individual reviews below, we regret to inform you that your work will not be considered further for publication in *eLife*.

The reviewers each highlighted the general importance of the work and the notion that this work may be revealing an important new angle on aminoglycoside mechanism of action. However, there were significant concerns raised about whether the calcium transients were simply correlated with cell death or truly causal. And precisely how calcium transients would, at a mechanistic level, result in cell death was unclear. These concerns are laid out in some detail in the individual reviews provided below and we hope they will be helpful to the authors.

Reviewer #1:

This paper tackles an important and interesting problem, namely why aminoglycosides are bactericidal. Relying largely on a GCaMP6 reporter that is a calcium sensor, the authors provide some data to support a claim that aminoglycosides trigger hyperpolarization of the cell, which is ultimately lethal. Although this represents an intriguing new angle on aminoglycoside mechanism of action, I thought some of the results were overinterpreted and at the end of the paper I'm still left wondering (i) how aminoglycosides lead to the purported increase in ATP that the authors think drives reverse action of the F1Fo ATPase to produce hyperpolarization and (ii) how hyperpolarization ultimately kills cells.

1) Figure 2D: It definitely seems like low pH reduces transients, but the authors should verify that there is, in fact, less killing by aminoglycosides in this low pH environment.

4) For the results in Figure 3E, the authors argue that there's no correlation between GTTR uptake and calcium transients, but there seems to be some sort of pattern in which GTTR uptake increases after a long burst of transients, i.e. toward the end of each trace shown. So, I'm not fully convinced there's no relationship.

3) "However, single cell data showed the initiation of transients and the uptake of PI were not correlated. The catastrophic transients started before dye uptake in all observed cells." But couldn't it be that catastrophic transients arise more easily than PI uptake but that both are reporting on aminoglycoside-induced membrane defects?

4) I guess I don't really understand why pH and protonophore CCCP both diminish transients and aminoglycoside efficacy. Do these treatments prevent hyperpolarization or aminoglycoside uptake or something else? I can see why CCCP likely prevents hyperpolarization by dissipating the PMF, but then shouldn't acidic pH conditions exacerbate not rescue the effects seen?

5) The increase in CFU in Figure 4E following addition of CCCP at 1 hr is not particularly large – it looks like an increase of only 2-3 fold. Given the central importance of this experiment to the model, I'm worried about overinterpretation of modest effect sizes here.

6) In Figure 5 I think it's essential for the authors to measure ATP concentration in a more direct way than their fluorescent reporter, which doesn't seem to have a particularly large dynamic range.

7) Why do large calcium transients kill cells?

8) I don't really understand why aminoglycoside increase ATP levels as hypothesized based on the results in Figure 5. If it's just a matter of inhibiting translation, then shouldn't a bacteriostatic antibiotic like chloramphenicol show a similar increase in ATP?

Reviewer #2:

In the manuscript by Bruni et al., from Kralj's lab, the authors proposed that aminoglycoside bactericidal action arises from dysregulated membrane potential. The authors used a genetically encoded calcium sensor previously developed in Bruni et al., 2017.

The Introduction is very short and could contain more information about what is known on aminoglycoside uptake (EDP-I and II, PMF, feed forward loop,… Taber et al., 1987).

The results are interesting, provocative and conceptually new. However, clarifications, controls and new experiences are needed to support their conclusions.

Essential revisions:

My main concern is that the authors consider the uptake as an aminoglycoside mechanism of action. This point of view leads to surprising conclusions as in the title of the paper:

“Membrane voltage dysregulation, not uptake, underlies bactericidal activity of aminoglycosides.”

For me, aminoglycoside act on ribosomes, so uptake is essential, without uptake no death. The authors use a fluorescent gentamicin (GTTR) to track the uptake and observe that the calcium transients appear before the drug uptake. Is GTTR fluorescence detection sufficiently sensitive for this type of experiment?

It will be interesting if the authors could use their tools (GCaMP6, GTTR) in the conditions previously used in others studies to support their conclusions: adding chloramphenicol to block translation and EDP-II step; used PMF altered strains (*nuo* sdh, Fe-S biosynthesis) and Gm-resistant strain.

In the same vein, the authors need to clarify the impact of low pH, anoxic medium and CCCP (subsection “Aminoglycosides induced catastrophic calcium transients”), these three conditions are known to decrease PMF (essential for aminoglycoside uptake). Instead, the authors talk about "environments that diminish aminoglycoside efficacy".

The most surprising result is that even in presence of CCCP, the ribosomes are still dissociated with aminoglycoside, leading to the conclusion that voltage is not necessary for aminoglycoside uptake. Is it possible to detect GTTR fluorescence in the presence of CCCP?

In all experiments, the GCaMP6 fluorescence tends to increase even before adding drugs. Given that previous study of the same lab have shown that using agarose pad lead to a voltage-induced calcium flux via a local mechanical environment, it will be essential to represent in a same figure the control without aminoglycoside. Even though in the text (subsection “Aminoglycosides induced catastrophic calcium transients”) this comparison is made.

Reviewer #3:

Bruni and Kralj present data supporting that bactericidal activity of aminoglycosides relies on membrane voltage dysregulation. Specifically, the authors describe an interesting observation of fast calcium fluctuations in *E. coli*, under the addition of aminoglycosides that correlates with cellular death. Driven by this observation, they perform additional experiments and conclude that aminoglycosides increase ATP concentration inside bacterial cells, which reverses the F1Fo-ATPase activity, causing hyperpolarization and eventually cell death. Together, this work provides a better understanding of the bactericidal mechanism of antibiotics, which is very significant. In general, I found the paper meaningful, but I have major concerns regarding the main claims, and specifically regarding the proposed mechanism of action.

Essential revisions:

1) The authors claim that the increased cellular ATP reverses the activity of the F1Fo-ATPase generating a hyperpolarized state of the cell. While I agree with the authors that reverse activity of F1Fo-ATPase can cause hyperpolarization, it is unclear to me if this is the only, or even the major mechanism of hyperpolarization upon aminoglycoside treatment. The authors present two lines of evidence to argue their point: ATPase knockout mutants and mgtC expression data.

My first concern with the author's claims is that while some ATPase knockouts show a reduction of aminoglycosides action, there is no single mutant that abolishes the effect (changes in mean SD of calcium transient or recover CFUs). This is in contrast to the results where the authors truly abolish the hyperpolarization through other means such as CCCP or pH. This discrepancy suggests the possibility that the reverse activity of F1Fo-ATPase may not by the sole cause, or even the major cause of the hyperpolarization.

Second, the expression of MgtC eliminates the bactericidal activity of aminoglycosides. MgtC is a magnesium ion transporter, and thus it is also plausible to think that increased magnesium ion uptake prevents hyperpolarization.

2) Contrary to the authors claim, I could not find evidence in the manuscript to support that the catastrophic calcium transients are the direct cause of cell death. Even more, the fact that any cell that experienced calcium transients in the untreated condition did not divide, but did also not die (Introduction), provokes the question whether calcium transients are the true cause of death.

The authors would be better off, stating the relationship between calcium transients and cell death as a correlation rather than causation. However, if the authors choose to stick with their claim of causality, they have to show that calcium transients itself are sufficient to kill the cell, regardless of potential other effects related with hyperpolarization. Perhaps the authors could use a specific calcium chelator, such as BAPTA, to prevent calcium transient and therefore cellular death?

Below I detail my major concerns regarding the claimed mechanism of action:

- One of the most important results in this manuscript is that some ATPase knockouts affect calcium transients. To show the change between these mutants and WT strain, the authors measured differences in the mean GCaMP6 SD signal. My problem is that the measurements for different knockouts were done at different time points (Figure 5D top and bottom. See Table 1). I don't think that this is the proper way of comparing strains. It is unclear why the authors chose this strange comparison.

- Why is there so much variation of the GCaMP6 signal between WT replicas? For example, the same Figure 5D, and Figure 2—figure supplement 4C, Supplementary igure 5A, and 5E. In some cases, the variation between different WT replicas could be even larger than the difference between ATPase mutants and the WT strain.

- Furthermore, shouldn't the ATP concentration decrease if F1Fo-ATPase keeps hydrolyzing ATP?

- It is also not clear why the atpC knockout should have a higher membrane potential than WT. In the discussion, the authors do not mention anything about atpC.

- The authors do not mention why calcium transients (and possibly increase in ATP concentration) are not observed in other antibiotics that also impair ribosomal activity, such as chloramphenicol?

- Importantly, GCaMP6 signal doesn't report directly on the membrane potential. Therefore, to determine whether hyperpolarization induces a positive feedback on gentamycin uptake, it would be necessary to use a more direct membrane potential reporter, such as the one the authors used for their other measurements.

[Editors’ note: further revisions were suggested prior to acceptance, as described below.]

Thank you for submitting your article "Membrane voltage dysregulation driven by metabolic dysfunction underlies bactericidal activity of aminoglycosides" for consideration by *eLife*. Your article has been reviewed by three peer reviewers, one of whom is a member of our Board of Reviewing Editors, and the evaluation has been overseen by Wendy Garrett as the Senior Editor. The following individuals involved in review of your submission have agreed to reveal their identity: Daisy Lee (Reviewer #3).

The reviewers have discussed the reviews with one another and the Reviewing Editor has drafted this decision to help you prepare a revised submission.

This paper examines the mechanism of killing by aminoglycoside antibiotics, providing evidence that their bactericidal activity stems from a hyperpolarization of the inner membrane of cells rather than affecting uptake, or at least rather than affecting only uptake. This revised version of the manuscript was deemed substantially improved by the reviewers and they are each enthusiastic about the work and the prospect of publishing it in *eLife*. However, one of the reviewers raised a couple of important points about the role of ATP and the reversal of ATP synthase that the authors should address, either by providing additional data or by adjusting the language in the papers and the claims made.

1) The revised manuscript still does not include an experiment addressing aminoglycoside uptake. Did the authors have any alternative to replace GTTR?

2) It is unclear what 'by metabolic dysfunction' in the title means. Do the authors mean increased intracellular ATP? While the abstract claims that 'the hyperpolarization arose from altered ATP flux', it is hard to understand why or how F1Fo ATPase reverts its action only when the membrane potential is high enough (inside is more negative), not in the other way around (inside is not as negative – it seems to me it's better to revert the action in this case since then they can increase the membrane potential to the normal level through reverting it…).

3) Throughout the data, it seems that the membrane potential plays a critical role in exerting bactericidal effects. While ATPase mutants and the mgtC expressing strain presented here likely have altered membrane potential without any antibiotic treatment, I couldn't find data showing their basal membrane potential level compared to the WT. Without the basal membrane potential data, it is impossible to discern if the phenotypes of the mutants are due to their basal membrane potential differences or by the author's main claim in the abstract. It would be very valuable if the authors can provide the basal membrane potential level of mutants compared to WT, CCCP added case, and/or pH6.5 case in the study.

---

## [Author Response]

[Editors’ note: the authors resubmitted a revised version of the paper for consideration. What follows is the authors’ response to the first round of review.]

[…]Reviewer #1:This paper tackles an important and interesting problem, namely why aminoglycosides are bactericidal. Relying largely on a GCaMP6 reporter that is a calcium sensor, the authors provide some data to support a claim that aminoglycosides trigger hyperpolarization of the cell, which is ultimately lethal. Although this represents an intriguing new angle on aminoglycoside mechanism of action, I thought some of the results were overinterpreted and at the end of the paper I'm still left wondering (i) how aminoglycosides lead to the purported increase in ATP that the authors think drives reverse action of the F1Fo ATPase to produce hyperpolarization and (ii) how hyperpolarization ultimately kills cells.

We regret having overinterpreted some of the results that we presented. We have reassessed many of our results in the context of ample previous and current evidence related to aminoglycoside induced cell death and attempted to put our results in context of the broader picture of our collective understanding of aminoglycoside’s efficacy.

We have addressed two of the points this reviewer had been left wondering about with the following:

i) We have included a more detailed contextual description of how we think aminoglycosides lead to the altered ATP flux in subsection “ATP dysregulation precedes voltage induced bactericidal killing”.

ii)A more detailed description of how hyperpolarization might be killing cells is included in the Discussion section.

1) Figure 2D: It definitely seems like low pH reduces transients, but the authors should verify that there is, in fact, less killing by aminoglycosides in this low pH environment.

There is evidence of low pH protection in the field (Taber et al., 1987; Davis, 1987; Damper and Epstein, 1981). Unfortunately, we did not have time to address this prior to the COVID related loss of lab activity. We have some data in the lab which shows single cell data correlated with continued growth for up to 16 hours at low pH in the presence of aminoglycoside that we’re happy to share to provide some evidence that growth is still possible, but we do not have the gold standard CFU assay to back it up, and will not until we return to lab.

2) For the results in Figure 3E, the authors argue that there's no correlation between GTTR uptake and calcium transients, but there seems to be some sort of pattern in which GTTR uptake increases after a long burst of transients, i.e. toward the end of each trace shown. So, I'm not fully convinced there's no relationship.

We agree that the correlation in time between GTTR uptake and calcium transients being sustained argues that there could be a relationship between the two. Some data that opposes this positive correlation is in the Materials and methods section Figure 1D, in combination with Figure 4A, B. The presence of CCCP eliminates transients but still allows large fluorescent molecules (like GTTR) to accumulate due to the pore formation aminoglycosides create. We instead focus on propidium iodide as a means to measure membrane breakdown.

3) "However, single cell data showed the initiation of transients and the uptake of PI were not correlated. The catastrophic transients started before dye uptake in all observed cells." But couldn't it be that catastrophic transients arise more easily than PI uptake but that both are reporting on aminoglycoside-induced membrane defects?

Yes. You’re absolutely right. See the Discussion section.

4) I guess I don't really understand why pH and protonophore CCCP both diminish transients and aminoglycoside efficacy. Do these treatments prevent hyperpolarization or aminoglycoside uptake or something else? I can see why CCCP likely prevents hyperpolarization by dissipating the PMF, but then shouldn't acidic pH conditions exacerbate not rescue the effects seen?

PMF is comprised of both voltage and pH differences, so as the external pH is lowered, more of the PMF is carried in the pH term, meaning that less voltage is needed. At pH 6, cells need no voltage in order to maintain ~100 mV of PMF. We have included a more detailed description in the Materials and methods section.

5) The increase in CFU in Figure 4E following addition of CCCP at 1 hr is not particularly large – it looks like an increase of only 2-3 fold. Given the central importance of this experiment to the model, I'm worried about overinterpretation of modest effect sizes here.

The axis was in log scale, but we apologize for not making the increase more apparent in the text. Please see subsection “Voltage toggles between bactericidal and bacteriostatic activity in aminoglycoside treated cells”.

6) In Figure 5 I think it's essential for the authors to measure ATP concentration in a more direct way than their fluorescent reporter, which doesn't seem to have a particularly large dynamic range.

We agree that the iATPSnFR does not have a large dynamic range and in response we did undertake new experiments to test absolute changes. In light of this new experiment, which was contradictory to the iATPSnFR data, we reformulated our explanation and found many pieces of literature that substantiated this new explanation. Please see Materials and methods section, Figure 5—figure supplement 1D, and subsection “ATP dysregulation precedes voltage induced bactericidal killing”.

7) Why do large calcium transients kill cells?

We are unable to say definitively that the calcium transients prevent future cell division, though this remains an intriguing possibility. In the current draft, we are much more specific in saying that we use the calcium transients as means to visualize death, but cannot prove they induce death. We have multiple instances reinforcing that there is no evidence that large calcium transients kill cells including subsection “Single cell calcium flux predicts cellular aminoglycoside response”.

8) I don't really understand why aminoglycoside increase ATP levels as hypothesized based on the results in Figure 5. If it's just a matter of inhibiting translation, then shouldn't a bacteriostatic antibiotic like chloramphenicol show a similar increase in ATP?

Yes. And some non-aminoglycoside translation inhibitors do have ATP increases with identical kinetics while others show a significantly different kinetics but ultimately higher fluorescence levels (the data wasn’t shown out to that time point to simplify figure presentation), Materials and methods section, Figure 5—figure supplement 1C.

Reviewer #2:In the manuscript by Bruni et al., from Kralj's lab, the authors proposed that aminoglycoside bactericidal action arises from dysregulated membrane potential. The authors used a genetically encoded calcium sensor previously developed in Bruni et al., 2017.The Introduction is very short and could contain more information about what is known on aminoglycoside uptake (EDP-I and II, PMF, feed forward loop,… Taber et al., 1987).The results are interesting, provocative and conceptually new. However, clarifications, controls and new experiences are needed to support their conclusions.

We thank the reviewer for the excellent references and idea to improve our introduction. We have heeded this advice and overhauled the introduction in favor of a historically relevant context for our results. This has also altered some of the previously overstated conclusions we drew and hope that this new draft is more succinct and clearer.

Essential revisions:My main concern is that the authors consider the uptake as an aminoglycoside mechanism of action. This point of view leads to surprising conclusions as in the title of the paper:“Membrane voltage dysregulation, not uptake, underlies bactericidal activity of aminoglycosides.”

We have altered the title and introduction to better reflect our actual observations.

For me, aminoglycoside act on ribosomes, so uptake is essential, without uptake no death. The authors use a fluorescent gentamicin (GTTR) to track the uptake and observe that the calcium transients appear before the drug uptake. Is GTTR fluorescence detection sufficiently sensitive for this type of experiment?

We agree that uptake is essential. We have provided evidence that uptake sufficient to kill the cells in the absence of a membrane potential (determined after membrane potential was restored in the absence of aminoglycoside) can occur in these *E. coli*. Main text figure 1C combined with Figure 4C, D

It will be interesting if the authors could use their tools (GCaMP6, GTTR) in the conditions previously used in others studies to support their conclusions: adding chloramphenicol to block translation and EDP-II step; used PMF altered strains (nuo sdh, Fe-S biosynthesis) and Gm-resistant strain.

We have included some of our ∆*nuo* experiments throughout the new text (Figure 2—figure supplement 1B). Sadly, some of the other experiments proposed were not reached prior to us leaving the lab, but we look forward to conducting them in a follow-up study. The reviewer should feel free to contact us without revealing their identity if they’d like to discuss or collaborate on these types of experiments further, as we think there is ample space to improve the current aminoglycoside death model with our new tools. Furthermore, the GTTR data became suspect when we explored our PI data further. If uptake up PI can occur in a majority of cells, even in the absence of a membrane potential, then the uptake of GTTR cannot be considered to be occurring through the typical uptake route of aminoglycoside because they can enter through pores that are not there during the initial aminoglycoside uptake. We have removed the GTTR data as a result of these conclusions.

In the same vein, the authors need to clarify the impact of low pH, anoxic medium and CCCP (subsection “Aminoglycosides induced catastrophic calcium transients”), these three conditions are known to decrease PMF (essential for aminoglycoside uptake). Instead, the authors talk about "environments that diminish aminoglycoside efficacy".

We have clarified the impact of low pH and CCCP and attempted to better explain their inhibitory effects in the supplementary discussion (Materials and methods section). We have removed the anoxic medium piece of data because we have been unable to set up anoxic conditions for all of the necessary experiments to clearly argue their protective effect throughout the text. We hope to address the differences anoxic conditions create in a future publication as we believe it is more nuanced than a decreased PMF.

The most surprising result is that even in presence of CCCP, the ribosomes are still dissociated with aminoglycoside, leading to the conclusion that voltage is not necessary for aminoglycoside uptake. Is it possible to detect GTTR fluorescence in the presence of CCCP?

This surprised us as well, and ultimately inspired a good portion of this project’s direction. We have altered the main text to highlight this point, including the ribosome dissociation in the first figure. We have removed the GTTR data because of the Propidium iodide data in main text Figure 1D: GTTR increased on the same timescale as our PI experiments, so we cannot rule out GTTR uptake through pores. Therefore, the GTTR data is suspect unless it can be detected prior to PI uptake (spectral overlap of GTTR and PI preclude us from determining this result). We have altered the text to reflect this.

In all experiments, the GCaMP6 fluorescence tends to increase even before adding drugs. Given that previous study of the same lab have shown that using agarose pad lead to a voltage-induced calcium flux via a local mechanical environment, it will be essential to represent in a same figure the control without aminoglycoside. Even though in the text (subsection “Aminoglycosides induced catastrophic calcium transients”) this comparison is made.

We have included main text Figure 2A, C to address this shortcoming.

Reviewer #3:[…]Essential revisions:1) The authors claim that the increased cellular ATP reverses the activity of the F1Fo-ATPase generating a hyperpolarized state of the cell. While I agree with the authors that reverse activity of F1Fo-ATPase can cause hyperpolarization, it is unclear to me if this is the only, or even the major mechanism of hyperpolarization upon aminoglycoside treatment. The authors present two lines of evidence to argue their point: ATPase knockout mutants and mgtC expression data.

We agree that there are numerous ATPases that can affect membrane voltage, which will temper the potential importance of the F-ATPase in the context of aminoglycoside killing. Hopefully our revisions to the text have clarified that this molecular machine is merely one, albeit significant and important, contributor to aminoglycoside killing in this context. We have found that in strain with all components of the ATPase, there are still a few transients, though much reduced compared to the parent strain (Figure 5E).

My first concern with the author's claims is that while some ATPase knockouts show a reduction of aminoglycosides action, there is no single mutant that abolishes the effect (changes in mean SD of calcium transient or recover CFUs). This is in contrast to the results where the authors truly abolish the hyperpolarization through other means such as CCCP or pH. This discrepancy suggests the possibility that the reverse activity of F1Fo-ATPase may not by the sole cause, or even the major cause of the hyperpolarization.

We found the observation that no single mutant abolished the aminoglycoside effect unsettling as well and further reflected on this point. In order to address this concern, we have included a complete knockout of the F-ATPase (DK8) in main text Figure 5E. Unfortunately, due to COVID related loss of laboratory access the corresponding DK8 CFU experiments remain to be completed. Regardless of that outcome, we have adapted the text by downplaying what we initially argued was a central role for this ATPase. Instead, we now invoke a relatively well-established translation inhibitor cellular response – the dysregulation of metabolism – as the main driver of cell death. While the F-ATPase contributes to this cell death through hyperpolarization, it is not exclusively responsible given the single component knockout results.

Second, the expression of MgtC eliminates the bactericidal activity of aminoglycosides. MgtC is a magnesium ion transporter, and thus it is also plausible to think that increased magnesium ion uptake prevents hyperpolarization.

While mgtC is upregulated in response to low magnesium and aids in Mg uptake with other components of its operon, we are unaware of evidence that mgtC alone is a magnesium ion transporter. There is even evidence that the *M. tuberculosis* mgtC does not bind magnesium https://www.ncbi.nlm.nih.gov/pubmed/22984256, and further most of the evidence we have found for the *Salmonella* mgtC function appears to be one of a regulatory role in the cell. If we have grossly missed evidence to the contrary, please let us know what that is so that we can run additional experiments to verify that the function of mgtC we claim is due to F-ATPase activity inhibition rather than Mg uptake.

2) Contrary to the authors claim, I could not find evidence in the manuscript to support that the catastrophic calcium transients are the direct cause of cell death. Even more, the fact that any cell that experienced calcium transients in the untreated condition did not divide, but did also not die (Introduction), provokes the question whether calcium transients are the true cause of death.The authors would be better off, stating the relationship between calcium transients and cell death as a correlation rather than causation. However, if the authors choose to stick with their claim of causality, they have to show that calcium transients itself are sufficient to kill the cell, regardless of potential other effects related with hyperpolarization. Perhaps the authors could use a specific calcium chelator, such as BAPTA, to prevent calcium transient and therefore cellular death?

We cannot prove calcium transients are the direct cause of cell death. We unfortunately did have a sentence that insinuated this in the prior version of the text. That has been removed. We have multiple instances reinforcing that there is no evidence that large calcium transients kill cells including subsection “Single cell calcium flux predicts cellular aminoglycoside response”.

Below I detail my major concerns regarding the claimed mechanism of action:- One of the most important results in this manuscript is that some ATPase knockouts affect calcium transients. To show the change between these mutants and WT strain, the authors measured differences in the mean GCaMP6 SD signal. My problem is that the measurements for different knockouts were done at different time points (Figure 5D top and bottom. See Table 1). I don't think that this is the proper way of comparing strains. It is unclear why the authors chose this strange comparison.

We believe that the majority of these ATPase knockouts are affecting the onset of transients due to their differential effects on the polarization state of the cell, and therefore would exhibit altered kinetics. We are happy to run statistical testing at multiple time points if that is preferable. We can also separate different strain’s GCSD traces in the figure to improve readability if this was the main cause of concern.

- Why is there so much variation of the GCaMP6 signal between WT replicas? For example, the same Figure 5D, and Figure2—figure supplement 4EC, Supplementary figure 5A, and 5E. In some cases, the variation between different WT replicas could be even larger than the difference between ATPase mutants and the WT strain.

The standard deviation metrics will include aspects of the signal-to-noise characteristics of the GCaMP images, which are intrinsically tied to the microscope used to collect data. During our experiments, we used more than one microscope (one with a sCMOS camera and 40x air objective, and one with a EMCCD with a 100x oil objective) which have very different noise characteristics. Thus, we believe the only way to view this type of analysis is through a relative measurement of untreated or wildtype cells taken at the same time. For all data shown, the biological replicates and conditions were taken on the same day on the same microscope and used for comparison. Using these relative metrics, we can easily identify treated vs untreated conditions, as well as investigate the kinetics and amplitude of the calcium transients.

- Furthermore, shouldn't the ATP concentration decrease if F1Fo-ATPase keeps hydrolyzing ATP?

On the basis of newly included experiments, we have adapted our thinking of the ATP result. The steady state level of ATP changes very little, but the flux through non-ribosomal components increases dramatically. Please see Materials and methods section, Figure 5—figure supplement 1D, and subsection “ATP dysregulation precedes voltage induced bactericidal killing”.

- It is also not clear why the atpC knockout should have a higher membrane potential than WT. In the Discussion section, the authors do not mention anything about atpC.

Please see subsection “ATP dysregulation precedes voltage induced bactericidal killing”. We have added a brief discussion on about the roles of atpH and atpC. Discussion section.

- The authors do not mention why calcium transients (and possibly increase in ATP concentration) are not observed in other antibiotics that also impair ribosomal activity, such as chloramphenicol?

Additional experiments have confirmed that non-aminoglycoside translation inhibitors do increase ATP flux, and some even alter basal membrane potential (data not included). This new data supports a model that incorporates previously known aminoglycoside effects including mistranslated proteins and membrane pores helping to explain the difference in bactericidal activity. Hopefully this discrepancy is now clear. Also see Figure 5—figure supplement 1C.

- Importantly, GCaMP6 signal doesn't report directly on the membrane potential. Therefore, to determine whether hyperpolarization induces a positive feedback on gentamycin uptake, it would be necessary to use a more direct membrane potential reporter, such as the one the authors used for their other measurements.

We agree that the hyperpolarization could induce a positive feedback on gentamycin uptake. However, we see that even in the absence of membrane potential, enough aminoglycoside enters the cell to dissociate the ribosome and increase ATP. The hyperpolarization is required for bactericidal activity, though, which is a new way of viewing the role of voltage in the activity of aminoglycosides.

[Editors’ note: what follows is the authors’ response to the second round of review.]

[…]1) The revised manuscript still does not include an experiment addressing aminoglycoside uptake. Did the authors have any alternative to replace GTTR?

We initially removed the GTTR data from our revised manuscript since the ribosome dissociation data is more specific indicator of aminoglycoside activity. We have added it back in to help support our hypothesis. Figure 1—figure supplement figure 1D (a new panel) shows GTTR uptake in the absence or presence of CCCP, again showing dye uptake into cells regardless of membrane voltage. Furthermore, given our data with PI showing a large non-specific uptake, the GTTR signal may be confounded with our factors, which warrants caution in interpretation.

2) It is unclear what 'by metabolic dysfunction' in the title means. Do the authors mean increased intracellular ATP? While the abstract claims that 'the hyperpolarization arose from altered ATP flux', it is hard to understand why or how F1Fo ATPase reverts its action only when the membrane potential is high enough (inside is more negative), not in the other way around (inside is not as negative – it seems to me it's better to revert the action in this case since then they can increase the membrane potential to the normal level through reverting it…).

We apologize for confusion in the presentation of the model. When we refer to metabolic dysfunction, we are referring to deleterious usage of ATP within the cell. The F1Fo ATPase can trade negative (inside) voltage for ATP. So if it makes ATP, it is moving the voltage closer to 0 mV. On the other hand, the pump can also use ATP to increase (more negative) the membrane polarization. So if the balance of ATP to voltage is tilted, the motor can shift from consuming ATP to hydrolyzing ATP. In our model, ATP is not consumed by ribosomes, and is therefore free to be consumed by other ATPases (including the F1Fo ATPase) which will then hyperpolarize the membrane, leading to the transients.

We have attempted to clarify the model in Figure 5—figure supplement 3 (new supplementary figure) showing this stepwise process. Furthermore, we revised our thinking on calling step 2 “increased cellular ATP” in lieu of our BacTiter-Glo data, showing in bulk that cells do not have more ATP. Rather we believe that there is the same amount of ATP, however the metabolic burden of translation is no longer contributing to ATP consumption. Other groups have seen similar shifts in metabolic load in the presence of aminoglycosides and other translation inhibitors (cited in the relevant parts of the paper), so we believe this is part of a more universal effect of translation inhibitors causing metabolic dysfunction. Our language in subsection “ATP dysregulation precedes voltage induced bactericidal killing” of the main text document reflect this change in thinking, however if it is not clear that we mean a shift in ATP flux, rather than an increase in bulk ATP concentrations, we will revise our language to better reflect this thinking.

3) Throughout the data, it seems that the membrane potential plays a critical role in exerting bactericidal effects. While ATPase mutants and the mgtC expressing strain presented here likely have altered membrane potential without any antibiotic treatment, I couldn't find data showing their basal membrane potential level compared to the WT. Without the basal membrane potential data, it is impossible to discern if the phenotypes of the mutants are due to their basal membrane potential differences or by the author's main claim in the abstract. It would be very valuable if the authors can provide the basal membrane potential level of mutants compared to WT, CCCP added case, and/or pH6.5 case in the study.

We agree that membrane potential plays a critical role in the aminoglycoside response, and that the protection afforded by these mutations could be due to decreased voltage. To test the basal voltage, we have included a table and figure (Figure 5—figure supplement 2) which contains the membrane potentials for individual strains BW, DK8 (∆*unc* – F-ATPase components are all removed), and BW+mgtC. We observed that the DK8 and BW+mgtC strain had a more polarized basal membrane potential as compared to the BW strain alone. Finally, to see if this result was due to the strain, or some effect that is specific to the strain treated with TMRM, we utilized the less sensitive Nernstian dye DiOC_6_ to assess if dye uptake was altered in the DK8 strain overall. We observed results similar to our TMRM data with the dye DiOC_6_. These data are consistent with the interpretation that protection arises from altered ATP consumption, and not depolarization.